# Transfer Learning for Nonparametric Contextual Dynamic Pricing

**Fan Wang** [1]  **Feiyu Jiang** [2]  **Zifeng Zhao** [3]  **Yi Yu** [1]

## Abstract

Dynamic pricing strategies are crucial for firms to maximize revenue by adjusting prices based on market conditions and customer characteristics. However, designing optimal pricing strategies becomes challenging when historical data are limited, as is often the case when launching new products or entering new markets. One promising approach to overcome this limitation is to leverage information from related products or markets to inform the focal pricing decisions. In this paper, we explore transfer learning for nonparametric contextual dynamic pricing under a covariate shift model, where the marginal distributions of covariates differ between source and target domains while the reward functions remain the same. We propose a novel Transfer Learning for Dynamic Pricing (TLDP) algorithm that can effectively leverage pre-collected data from a source domain to enhance pricing decisions in the target domain. The regret upper bound of TLDP is established under a simple Lipschitz condition on the reward function. To establish the optimality of TLDP, we further derive a matching minimax lower bound, which includes the target-only scenario as a special case and is presented for the first time in the literature. Extensive numerical experiments validate our approach, demonstrating its superiority over existing methods and highlighting its practical utility in real-world applications.

## 1. Introduction

Dynamic pricing is a fundamental strategy used across many industries to maximize revenue by adjusting prices based on market conditions (e.g. Araman & Caldentey, 2009; Besbes & Zeevi, 2009; Wang et al., 2021b). In recent years, the rapid growth of online marketplaces and advances in data collection have further enabled sellers to use contextual information, such as customer characteristics, product features and market trends, to make more informed pricing decisions. This has fostered research in contextual dynamic pricing (e.g. Wang et al., 2023; Luo et al., 2024; Fan et al., 2024), which further incorporates in-depth information to optimize pricing strategies.

Effective dynamic pricing involves balancing the exploration of the unknown revenue model with the exploitation of the estimated revenue model to maximize rewards. The revenue model, representing the relationship between covariates, prices and revenue, has been explored under both parametric and nonparametric settings in the literature. The parametric settings impose specific assumptions on the revenue model and have been widely studied (e.g. Qiang & Bayati, 2016; Javanmard & Nazerzadeh, 2019; Ban & Keskin, 2021; Wang et al., 2021a; Zhao et al., 2024). The nonparametric settings provide more flexibility, making them more suitable for real-world applications where customer behaviour and market conditions are diverse and unpredictable (e.g. Chen & Gallego, 2021; Chen et al., 2023).

Despite the wide applications of contextual dynamic pricing, collecting sufficient data to design an optimal pricing strategy can be challenging, particularly when launching a new product or entering a new market with limited historical data. In contrast, abundant data may be available from related products or markets, from which one may leverage information to achieve improved decision-making. For instance, historical data from other platforms or existing markets can help sellers optimize pricing strategies more efficiently when entering new environments. This naturally falls in the territory of transfer learning (e.g. Pan & Yang, 2009), where datasets from similar but different distributions are utilized to enhance learning of a target dataset.

Transfer learning, as a research area, has been extensively studied in the machine learning literature, with applications in areas such as recommendation systems (e.g. Pan et al., 2010), language models (e.g. Han et al., 2021) and disease detection (e.g. Maqsood et al., 2019). It has recently also gained attention in statistics and has been explored in various problems, such as nonparametric regression (Cai & Pu, 2022b), contextual multi-armed bandits (Suk & Kpotufe,

---

[1]Department of Statistics, University of Warwick [2]School of Management, Fudan University [3]Mendoza College of Business, University of Notre Dame. Correspondence to: Fan Wang <fan.wang.6@warwick.ac.uk>.

*Proceedings of the 42$^{nd}$ International Conference on Machine Learning*, Vancouver, Canada. PMLR 267, 2025. Copyright 2025 by the author(s).

2021; Cai et al., 2024a) and functional data analysis (Cai et al., 2024b). In statistical transfer learning, two common settings are posterior drift, where the conditional distributions of responses given covariates vary (e.g. Reeve et al., 2021; Maity et al., 2022), and covariate shift, where the conditional distributions remain consistent but the marginal distributions of covariates differ (e.g. Hanneke & Kpotufe, 2019; Kpotufe & Martinet, 2021).

### 1.1. List of contributions

In this paper, we study transfer learning for nonparametric contextual dynamic pricing under the covariate shift model. The main contributions of this paper are summarized as follows.

Firstly, to the best of our knowledge, this is the first study to explore transfer learning in dynamic pricing. We introduce a novel algorithm, the Transfer Learning for Dynamic Pricing (TLDP) algorithm, which effectively leverages source domain data to enhance pricing decisions in the target domain.

Secondly, we establish theoretical guarantees for the proposed transfer learning algorithm and further derive a minimax lower bound. We demonstrate that the proposed algorithm achieves minimax optimal regret, with the cumulative regret (up to logarithmic factors) given by

$$n_Q\{n_Q + (\kappa n_P)^{\frac{d+3}{d+3+\gamma}}\}^{-\frac{1}{d+3}},$$

where $n_Q \in \mathbb{Z}_+$ represents the length of the target horizon, $n_P \in \mathbb{N}$ represents the number of observations in the pre-collected source dataset, $\gamma \in [0, \infty]$ (see Definition 2.3 later) denotes the transfer exponent quantifying the similarity between source and target covariate distributions and $\kappa \in [0, 1]$ (see Definition 2.4 later) represents the exploration coefficient quantifying the minimal level to which the source data adequately explores the price space.

Thirdly, as an important byproduct, our study covers the special case of nonparametric contextual dynamic pricing without transfer learning under the setting of a Lipschitz reward function. Our proposed algorithm achieves the regret upper bound of order

$$n_Q^{(d+2)/(d+3)}$$

up to logarithmic factors, with an accompanying minimax lower bound. To the best of our knowledge, this is the first minimax lower bound result under the Lipschitz condition for nonparametric contextual dynamic pricing with a general dimension $d \geq 1$.

Finally, we validate the proposed approach through extensive numerical experiments, demonstrating its effectiveness compared to existing methods and highlighting its practical utility in real-world applications.

### 1.2. Notation and organization

Denote $\mathcal{X} = [0, 1]^d$ as the covariate space, $\mathcal{P} = [0, 1]$ as the price space and $\mathcal{Z} = \mathcal{X} \times \mathcal{P}$ as the joint space of covariates and prices. *All* balls in this paper are defined with respect to the $\ell_\infty$-norm. Let $B(s, r)$, $B_\mathcal{X}(s, r)$ and $B_\mathcal{P}(s, r)$ denote the balls in $\mathcal{Z}$, $\mathcal{X}$ and $\mathcal{P}$, respectively, centred at $s$ with radius $r$. For any ball $B$, let $r(B)$ denote its radius and $c(B)$ its centre. For $n \in \mathbb{Z}_+$, let $[n] = \{1, \ldots, n\}$. For any distribution $P$, let $\mathrm{supp}(P)$ be its support.

The remainder of the paper is organized as follows. In Section 2, we formally define the problem. Section 3 introduces the TLDP algorithm. In Section 4, we present theoretical results on the regret bounds of the proposed method, followed by a matching minimax lower bound. We demonstrate the practical performance of our approach through comprehensive numerical experiments in Section 5 and conclude in Section 6.

## 2. Problem formulation

In this section, we introduce a nonparametric contextual dynamic pricing model under the transfer learning framework. Our goal is to minimize the regret for the target data by leveraging pre-collected data from a related source domain.

For the target domain, the seller sells a product to $n_Q \in \mathbb{Z}_+$ sequentially arriving consumers. At each time $t \in [n_Q]$, the seller observes contextual information $X_t \in \mathcal{X}$ drawn from the distribution $Q_X$. Based on $X_t$, the seller sets a price $p_t \in \mathcal{P}$ and then receives a random revenue $Y_t \in [0, 1]$ with its conditional expectation

$$\mathbb{E}\{Y_t | X_t, p_t\} = f(X_t, p_t), \tag{1}$$

where $f : \mathcal{Z} \to [0, 1]$ is the unknown reward function.

In the context of transfer learning, we further assume that the seller has access to a pre-collected source dataset

$$\mathcal{D}^P = \{(X_t^P, p_t^P, Y_t^P)\}_{t=1}^{n_P} \subset \mathcal{Z} \times [0, 1], \tag{2}$$

where $X_t^P \in \mathcal{X}$ is drawn from a distribution $P_X$, $p_t^P \in \mathcal{P}$ is the price and $Y_t^P$ corresponds to the observed random revenue with its conditional expectation $\mathbb{E}\{Y_t^P | X_t^P, p_t^P\} = f^P(X_t^P, p_t^P)$. As mentioned in the introduction, this paper focuses on the covariate shift model, where the marginal distributions of covariates between the source and target domains are different, i.e. $P_X \neq Q_X$, but the conditional distributions of rewards are identical, i.e.

$$f(x, p) = f^P(x, p), \quad \forall (x, p) \in [0, 1]^d \times [0, 1]. \tag{3}$$

Our primary objective is to design a pricing strategy that minimizes the cumulative regret over the selling horizon in the target domain by effectively utilizing both the source

data $\mathcal{D}^P$ and the observed target data. Formally, let $\Pi$ denote the family of all price policies $\pi = \{p_1^\pi, \ldots, p_{n_Q}^\pi\}$ where each price $p_t^\pi$ is $\mathcal{F}_{t-1}$-measurable, $t \in \mathbb{Z}_+$. The field $\mathcal{F}_{t-1}$ is the $\sigma$-algebra generated by the history target data $\{(X_s, p_s, Y_s)\}_{s=1}^{t-1}$, all source data $\{(X_s^P, p_s^P, y_s^P)\}_{s=1}^{n_P}$ and covariate $X_t$. For any price strategy $\pi \in \Pi$, the cumulative regret over $n_Q$ steps is defined as

$$R = R_\pi(n_Q) = \sum_{t=1}^{n_Q} \mathbb{E}\{f^*(X_t) - f(X_t, p_t^\pi)\}, \quad (4)$$

where $f^*(x) = \max_{p \in \mathcal{P}} f(x, p)$ is the maximum expected revenue for any $x \in \mathcal{X}$. We further impose the following assumptions on the reward function and the underlying distributions.

**Assumption 2.1.** Assume that the reward function $f \colon \mathcal{Z} \to [0, 1]$ is Lipschitz continuous with respect to the $\ell_\infty$-norm, i.e. there exists an absolute constant $C_{\text{Lip}} > 0$ such that for any $(x_1, p_1), (x_2, p_2) \in \mathcal{Z}$,

$$|f(x_1, p_1) - f(x_2, p_2)| \le C_{\text{Lip}} \|(x_1^\top, p_1)^\top - (x_2^\top, p_2)^\top\|_\infty.$$

Assumption 2.1 regulates the changes in the reward function led by the changes in covariates and prices. This is commonly used in the literature of online learning with nonparametric reward function (e.g. Slivkins, 2011; Chen & Gallego, 2021; Chen et al., 2023), facilitating theoretical analysis of the estimation and learning processes. Assumption 2.2 below is a regularity condition on $Q_X$ to ensure the covariate space will be explored sufficiently.

**Assumption 2.2.** Assume that for the target distribution $Q_X$, there exist constants $0 < c_Q < C_Q$ such that for any $x \in \text{supp}(Q_X)$ and $r \in (0, 1]$,

$$c_Q r^d \le Q_X(B_{\mathcal{X}}(x, r)) \le C_Q r^d.$$

To quantify the potential benefit of transfer learning from the source domain to the target domain, we introduce the notion of transfer exponent, a parameter widely used in the transfer learning literature with covariate shift (e.g. Kpotufe & Martinet, 2021; Suk & Kpotufe, 2021; Cai et al., 2024a). It quantifies the degree of similarity between the source and target covariate distributions, regulating the potential of effective knowledge transfer.

**Definition 2.3** (Transfer exponent). The transfer exponent $\gamma \in [0, \infty]$ of the source covariate distribution $P_X$ with respect to the target covariate distribution $Q_X$ is defined as

$$\gamma = \inf\Big\{\gamma' \ge 0 \Big| \exists \text{ a constant } 0 < c_{\gamma'} \le 1 \text{ such that}$$
$$P_X(B_{\mathcal{X}}(x, r)) \ge c_{\gamma'} r^{\gamma'} Q_X(B_{\mathcal{X}}(x, r)),$$
$$\forall x \in \text{supp}(Q_X), r \in (0, 1]\Big\}.$$

The transfer exponent $\gamma$ quantifies the extent to which the source covariate distribution $P_X$ covers the target covariate distribution $Q_X$. A smaller $\gamma$ indicates a greater overlap, enabling more efficient information transfer. In an extreme case where $P_X$ and $Q_X$ are identical, $\gamma = 0$ indicating a perfect overlap. This parameter is crucial for analyzing the potential performance improvements achievable by transfer learning in our framework.

To ensure that the source data provides sufficient variability in prices across different covariates, we further introduce the exploration coefficient. This parameter quantifies the minimal extent to which the source data adequately explores the price space. It is essential for accurately estimating the reward function over various price levels.

**Definition 2.4** (Exploration coefficient). Let $\mu$ denote the joint distribution of the source covariate-price pairs and let $P_X$ be the marginal distribution of the source covariates. Define the exploration coefficient $\kappa \in [0, 1]$ as

$$\kappa = \inf_{\substack{x \in \text{supp}(P_X), \\ r \in (0, 1/2], \\ p \in [r, 1-r]}} \frac{\mu([p - r, p + r] \times B_{\mathcal{X}}(x, r))}{2r \cdot P_X(B_{\mathcal{X}}(x, r))}.$$

The exploration coefficient $\kappa$ measures the minimal conditional probability density of the price given the covariates. A larger $\kappa$ indicates that the source data provides better exploration of the price space for each covariate. This coefficient is crucial in assessing the usefulness of the source data for learning the reward function in the target domain. The definition of $\kappa$ extends the exploration coefficient introduced in Cai et al. (2024a) for multi-armed bandits (MAB) to continuous action spaces in dynamic pricing. To illustrate this concept, consider the following example.

*Example* 2.5. Suppose that the source dataset $\mathcal{D}^P$ is defined in (2) and the source prices are drawn independently of source covariates from a distribution with density $h_P$ given by $h_P(p) = \delta_i$ if $p \in (a_{i-1}, a_i]$, for $i \in [m]$, where $0 = a_0 < a_1 < \cdots < a_{m-1} < a_m = 1$ and $\delta_i > 0$ such that $\int_0^1 h_P(p)\,dp = 1$. This is a piecewise uniform distribution and the exploration coefficient is $\kappa = \min_{i \in [m]} \delta_i$, which holds for any source covariate distribution $P_X$.

## 3. Transfer learning algorithm

In this section, we introduce the Transfer Learning for Dynamic Pricing (TLDP) algorithm, detailed in Algorithm 1. TLDP borrows the idea of contextual zooming from Slivkins (2011) proposed for MAB problems with only target data, and addresses the unique challenge of leveraging source data to refine the exploration of the covariate-price space in the target domain. At a high level, TLDP is an upper confidence bound (UCB) type algorithm that handles the nonparametric setting through an adaptive partitioning strat-

**Algorithm 1** Transfer Learning for Dynamic Pricing (TLDP)

**Input:** horizon length $n_Q$, source dataset $\mathcal{D}^P$ and the smallest radius to explore $\tilde{r}$
**Initialize:** $t \leftarrow 1$, $\mathcal{A}_t \leftarrow \{\mathcal{Z}\}$, $n_t(B) \leftarrow n_B^P(\mathcal{D}^P)$, $\mathbf{re}_t(B) \leftarrow \mathbf{re}_B^P(\mathcal{D}^P)$, $\forall B \in \mathcal{A}_t$
**while** $t \leq n_Q$ **do**
   Observe $X_t$
   $\mathbf{rel}_t \leftarrow \{B \in \mathcal{A}_t : \exists p \in [0,1], (X_t, p) \in \mathbf{dom}(B, \mathcal{A}_t)\}$
   $B^{\mathbf{sel}} \leftarrow$ uniformly choose from $\arg\max_{B \in \mathbf{rel}_t} I_t(B)$
   $p_t \sim \text{Uniform}\{p : (X_t, p) \in \mathbf{dom}(B^{\mathbf{sel}}, \mathcal{A}_t)\}$
   $n_t^Q(B^{\mathbf{sel}}) \leftarrow n_t(B^{\mathbf{sel}}) - n_{B^{\mathbf{sel}}}^P(\mathcal{D}^P)$
   **while** $n_t^Q(B^{\mathbf{sel}}) \geq T_{B^{\mathbf{sel}}}^Q$ and $r(B^{\mathbf{sel}}) \geq 2\tilde{r}$ **do**
     $B' \leftarrow B\big((X_t, p_t), r(B^{\mathbf{sel}})/2\big)$
     $\mathcal{A}_t \leftarrow \mathcal{A}_t \cup \{B'\}$, $n_t(B') \leftarrow n_{B'}^P(\mathcal{D}^P)$
     $\mathbf{re}_t(B') \leftarrow \mathbf{re}_{B'}^P(\mathcal{D}^P)$, $B^{\mathbf{sel}} \leftarrow B'$
   **end while**
   **for** $B \in \mathcal{A}_t \backslash \{B^{\mathbf{sel}}\}$ **do**
     $n_{t+1}(B) \leftarrow n_t(B)$, $\mathbf{re}_{t+1}(B) \leftarrow \mathbf{re}_t(B)$
   **end for**
   $n_{t+1}(B^{\mathbf{sel}}) \leftarrow n_t(B^{\mathbf{sel}}) + 1$
   $\mathbf{re}_{t+1}(B^{\mathbf{sel}}) \leftarrow \mathbf{re}_t(B^{\mathbf{sel}}) + Y_t$, $t \leftarrow t + 1$
**end while**

egy. Unlike approaches relying on predefined partitions, TLDP sequentially refines partitions of the covariate-price space in response to observed data. The key components of TLDP are outlined following Algorithm 1.

**A head start by the source data.** At the core, the pricing is conducted by sequentially discretizing the continuous covariate-price space $\mathcal{Z}$. To be specific, this is to produce a collection of active $l_\infty$-balls in $\mathcal{Z}$, namely $\mathcal{A}_t$ at step $t \in [n_Q]$. The source data provide a head start for such partitioning, starting with the entire space $\mathcal{A}_1 = \{\mathcal{Z}\}$. TLDP sequentially refines the partition and adds smaller balls into $\mathcal{A}_t$. This partitioning is to be detailed later. We first introduce some necessary notation. For a ball $B \subset \mathcal{Z}$, let its count and cumulative revenue utilizing the source dataset $\mathcal{D}^P$ be

$$n_B^P(\mathcal{D}^P) = \sum_{(X,p,Y) \in \mathcal{D}^P} \mathbb{1}\{(X, p) \in B\} \tag{5}$$

and

$$\mathbf{re}_B^P(\mathcal{D}^P) = \sum_{(X,p,Y) \in \mathcal{D}^P} Y \mathbb{1}\{(X, p) \in B\}. \tag{6}$$

For any ball $B \in \mathcal{A}_t$, let its domain be

$$\mathbf{dom}(B, \mathcal{A}_t) = B \backslash (\cup_{B' \in \mathcal{A}_t : r(B') < r(B)} B'), \tag{7}$$

excluding any overlaps caused by strictly smaller balls in $\mathcal{A}_t$, thereby preventing redundant coverage.

**Dynamic pricing via UCB.** At each time $t \in [n_Q]$, after observing the covariate $X_t$, TLDP sets a price by first selecting a ball $B$ in the covariate-price space from the current candidate set $\mathcal{A}_t$ that maximizes the revenue potential $I_t(B)$ - defined below, then randomly chooses a price such that the covariate-price pair $(X_t, p)$ is in $B$, to be specific, in $\mathbf{dom}(B, \mathcal{A}_t)$ as defined in (7).

*Definition of $I_t$.* Let $n_t(B)$ and $\mathbf{re}_t(B)$ denote the cumulative count and revenue of $B$ from the target data before time $t$ and the source data $\mathcal{D}^P$, defined as $n_t(B) = n_B^P(\mathcal{D}^P) + \sum_{s=1}^{t-1} \mathbb{1}\{(X_s, p_s) \in B\}$ and $\mathbf{re}_t(B) = \mathbf{re}_B^P(\mathcal{D}^P) + \sum_{s=1}^{t-1} Y_s \mathbb{1}\{(X_s, p_s) \in B\}$, with $n_B^P(\mathcal{D}^P)$ and $\mathbf{re}_B^P(B)$ defined in (5) and (6), respectively. For any $B \subset \mathcal{Z}$, let its UCB uncertainty level be

$$\mathbf{conf}_t(B) = 2\sqrt{\frac{\log\left\{n_Q \vee (\kappa n_P)^{\frac{d+3}{d+3+\gamma}}\right\}}{n_t(B)}}. \tag{8}$$

As $n_t(B)$ grows, i.e. more samples are collected in $B$, $\mathbf{conf}_t(B)$ decreases, reflecting increasing in confidence. The revenue potential index is then defined as

$$I_t(B) = C_I r(B) + \min_{B' \in \mathcal{A}_t} \big\{ I_t^{\mathbf{pre}}(B') \\ + C_I \|c(B) - c(B')\|_\infty \big\}, \tag{9}$$

where $C_I > 0$ is a constant and the pre-index $I_t^{\mathbf{pre}}(B)$ is given by

$$I_t^{\mathbf{pre}}(B) = v_t(B) + C_I r(B) + \mathbf{conf}_t(B), \tag{10}$$

with $v_t(B) = \mathbf{re}_t(B)/n_t(B)$.

The index $I_t(B)$ is adapted from its MAB counterpart proposed in Slivkins (2011) to balance the estimated reward $v_t(B)$, the radius of the ball $r(B)$, the uncertainty level $\mathbf{conf}_t(B)$ and its distance to its neighbours $\|c(B) - c(B')\|_\infty$ - the choice of the norm is reflected in the Lipschitz condition imposed in Assumption 2.1. The selection of inputs $\kappa$ and $C_I$ is further discussed in Section 5.1.

**Partitioning.** The last ingredient is the sequential partitioning. A ball is further refined when enough samples are collected and its radius is not too small. The refinement is conducted by adding a smaller sub-ball to the candidate set $\mathcal{A}_t$. To be specific, the partitioning is summoned when the target sample count $n_t^Q(B)$ exceeds $T_B^Q$, where

$$T_B^Q := T_B^Q(\mathcal{D}^P) = \begin{cases} 0, & \omega(B) < n_B^P(\mathcal{D}^P), \\ \omega(B), & \omega(B) \geq n_B^P(\mathcal{D}^P), \end{cases} \tag{11}$$

with

$$\omega(B) = \left\lceil \frac{\log\left\{n_Q \vee (\kappa n_P)^{\frac{d+3}{d+3+\gamma}}\right\}}{r(B)^2} \right\rceil. \tag{12}$$

This threshold incorporates the contributions from the source data. Note that when there is sufficient source data within the ball, no additional target data is required for exploration.

## 4. Minimax optimality

In this section, we establish theoretical guarantees for the TLDP algorithm presented in Algorithm 1. Specifically, we derive an upper bound on the cumulative regret and a matching minimax lower bound, demonstrating that TLDP achieves minimax optimality under the given assumptions. We also compare our findings with existing literature to highlight the contributions of our approach. We begin by presenting the upper bound on the regret achieved by TLDP.

**Theorem 4.1.** *Suppose that the source dataset $\mathcal{D}^P = \{(X_t^P, p_t^P, Y_t^P)\}_{t=1}^{n_P}$ is defined in (2) with triplets independent across time. Assume that the target dataset, defined in (1), satisfies (3), and that Assumptions 2.1 and 2.2 hold, with $C_I \geq C_{\mathrm{Lip}}$ where $C_I, C_{\mathrm{Lip}} > 0$ are constants defined in (9) and Assumption 2.1, respectively. Let Algorithm 1 have input $\tilde{r}$ satisfying that*

$$\tilde{r} = C_r \left[ \frac{\log\left\{ n_Q + (\kappa n_P)^{\frac{d+3}{d+3+\gamma}} \right\}}{n_Q + (\kappa n_P)^{\frac{d+3}{d+3+\gamma}}} \right]^{\frac{1}{d+3}}, \quad (13)$$

*and $C_r^4 c_\gamma c_Q \geq 8$, where $C_r > 0$ is a constant and $c_\gamma, c_Q > 0$ are constants defined in Definition 2.3 and Assumption 2.2, respectively. Let $\pi$ denote the TLDP policy given by Algorithm 1. Recall the cumulative regret $R$ defined in (4) and it holds that*

$$R \leq C n_Q \left\{ n_Q + (\kappa n_P)^{\frac{d+3}{d+3+\gamma}} \right\}^{-\frac{1}{d+3}}$$
$$\cdot \log^{\frac{1}{d+3}} \left\{ n_Q + (\kappa n_P)^{\frac{d+3}{d+3+\gamma}} \right\},$$

*where $C > 0$ is a constant only depending on constants $C_I$, $C_r$, $C_{\mathrm{Lip}}$, $c_\gamma$ and $c_Q$.*

*Remark* 4.2. Theorem 4.1 establishes the regret bound under the assumption that the reward function $f$ is Lipschitz continuous with respect to the $\ell_\infty$-norm (see Assumption 2.1). This assumption can be relaxed to accommodate more general smoothness conditions. In particular, one may consider a Hölder continuity assumption (e.g. Kleinberg, 2004; Cai & Pu, 2022a), where there exists an absolute constant $C_{\mathrm{h}} > 0$ such that for any $(x_1, p_1), (x_2, p_2) \in \mathcal{Z}$,

$$|f(x_1, p_1) - f(x_2, p_2)| \leq C_{\mathrm{h}} \|(x_1^\top, p_1)^\top - (x_2^\top, p_2)^\top\|_\infty^\beta,$$

for some smoothness parameter $\beta \in (0, 1]$. We believe that the techniques developed in this paper can be extended to this setting. Choosing the smallest exploration radius as

$$\tilde{r} = C_r \left[ \frac{\log\left\{ n_Q + (\kappa n_P)^{\frac{d+1+2\beta}{d+1+2\beta+\gamma}} \right\}}{n_Q + (\kappa n_P)^{\frac{d+1+2\beta}{d+1+2\beta+\gamma}}} \right]^{\frac{1}{d+1+2\beta}},$$

we conjecture the TLDP algorithm (Algorithm 1) can achieve a regret bound

$$R \lesssim n_Q \left\{ n_Q + (\kappa n_P)^{\frac{d+1+2\beta}{d+1+2\beta+\gamma}} \right\}^{-\frac{1}{d+1+2\beta}}$$
$$\cdot \log^{\frac{1}{d+1+2\beta}} \left\{ n_Q + (\kappa n_P)^{\frac{d+1+2\beta}{d+1+2\beta+\gamma}} \right\}.$$

The smallest radius to explore $\tilde{r}$ mediates an exploration-exploitation trade-off. A smaller $\tilde{r}$ leads to finer partitions and more precise local estimates but increases exploration cost, while a larger $\tilde{r}$ reduces the exploration cost at the expense of estimation accuracy. The choice of $\tilde{r}$ in (13) strikes a balance in this trade-off, ensuring the algorithm achieves optimal regret bounds.

The regret bound in Theorem 4.1 elucidates the dependence of the cumulative regret on the transfer exponent $\gamma$ and the exploration coefficient $\kappa$. A lower transfer exponent $\gamma$ indicates higher similarity between the source and target covariate distributions $P_X$ and $Q_X$, leading to a larger adjusted source data size $(\kappa n_P)^{(d+3)/(d+3+\gamma)}$ and thus reducing the regret.

*Remark* 4.3. As an alternative to the global exploration coefficient in Definition 2.4, one can instead define a scale-dependent exploration coefficient $\kappa_r \in [0, 1]$ for any radius $r \in (0, 1/2)$ as

$$\kappa_r = \inf_{\substack{x \in \mathrm{supp}(P_X), \\ r' \in [r, 1/2], \\ p \in [r', 1-r']}} \frac{\mu\left([p - r', p + r'] \times B_{\mathcal{X}}(x, r')\right)}{2r' \cdot P_X\left(B_{\mathcal{X}}(x, r')\right)}.$$

Note that $\kappa_r$ is non-decreasing in $r$ and satisfies that $\kappa = \inf_{r \in (0, 1/2)} \kappa_r$. It quantifies how well the source data explores the covariate-price space at a given scale. Under the conditions of Theorem 4.1, for any choice of the smallest radius to explore $\tilde{r} \in (0, 1/2]$, with the local coefficient $\kappa_{\tilde{r}}$, the regret satisfies that

$$R \lesssim \begin{cases} n_Q \tilde{r} + \tilde{r}^{-(d+2)} \log(n_Q), & n_Q \geq (\kappa_{\tilde{r}} n_P)^{\frac{d+3}{d+3+\gamma}}, \\ n_Q \tilde{r}, & n_Q < (\kappa_{\tilde{r}} n_P)^{\frac{d+3}{d+3+\gamma}}, \\ & \text{and } \tilde{r} \geq \left\{ \frac{8 \log\left\{ (\kappa_{\tilde{r}} n_P)^{\frac{d+3}{d+3+\gamma}} \right\}}{c_\gamma c_Q \kappa_{\tilde{r}} n_P} \right\}^{\frac{1}{d+3+\gamma}}, \\ \tilde{r}^{-(d+2)} \log(\kappa_{\tilde{r}} n_P), & \text{otherwise.} \end{cases}$$

These three regimes illustrate how both the sample size and the exploration radius $\tilde{r}$ jointly influence the overall regret. While we focus on the global coefficient $\kappa$ in this paper for simplicity, the local coefficient $\kappa_r$ characterizes tighter performance guarantees in scenarios where exploration quality varies across different scales, particularly useful when the source data do not uniformly cover the entire price space.

We now show that the regret upper bound in Theorem 4.1 is minimax optimal (up to a logarithmic factor) by presenting the following matching lower bound.

**Theorem 4.4.** *Let $\mathcal{I}(\gamma, c_\gamma, \kappa, C_{\text{Lip}}, c_Q)$ denote the class of nonparametric dynamic pricing problems such that $(i)$ the source dataset $\mathcal{D}^P = \{(X_t^P, p_t^P, Y_t^P)\}_{t=1}^{n_P}$, defined in (2), satisfies Definition 2.4 with triplets independent across time; $(ii)$ the target dataset, defined in (1), satisfies Assumption 2.2; and $(iii)$ both datasets satisfy (3), Assumption 2.1 and Definition 2.3. It holds that*

$$\inf_\pi \sup_{I \in \mathcal{I}(\gamma, c_\gamma, \kappa, C_{\text{Lip}}, c_Q)} R_{\pi, I}(n_Q)$$
$$\geq c n_Q \big(n_Q + (\kappa n_P)^{\frac{d+3}{d+3+\gamma}}\big)^{-\frac{1}{d+3}},$$

*where $c > 0$ is a constant only depending on constants $C_{\text{Lip}}$, $c_\gamma$ and $c_Q$.*

Theorems 4.1 and 4.4 together exhibit a phase transition depending on the relative sizes of the adjusted source data $(\kappa n_P)^{\frac{d+3}{d+3+\gamma}}$ and the target data $n_Q$. When $(\kappa n_P)^{\frac{d+3}{d+3+\gamma}} \gg n_Q$, the algorithm significantly benefits from transfer learning, otherwise the regret is of the same rate as when only the target data are used.

The most relevant work is Cai et al. (2024a), which studied transfer learning for nonparametric contextual MAB under covariate shift. In their setting, actions (i.e. arms) are discrete and for simplicity are treated as a constant (i.e. the number of arms $K \asymp 1$). They established a regret bound of order

$$n_Q\big(n_Q + (\kappa n_P)^{\frac{d+2}{d+2+\gamma}}\big)^{-\frac{\beta}{d+2\beta}}, \tag{14}$$

with $\beta$ denoting the smoothness parameter of the reward function ($\beta = 1$ in our setting). The simplification $K \asymp 1$ results in their regret bound losing explicit control of $K$, making their approach inapplicable to infinite or uncountable action spaces. In contrast, dynamic pricing presents a unique challenge due to its continuous action space (i.e. prices), requiring more intricate methodologies than that in Cai et al. (2024a). Our approach effectively addresses this by adaptively partitioning the joint covariate-price space. This adaptation ensures optimal regret scaling while accounting for both the covariates (of dimension $d$) and price (of dimension 1). Substituting $d + 1$ for $d$ in (14) shows that their regret bound aligns with the one established in Theorem 4.1.

When only target data are utilized, several existing studies have developed methods and analyzed regret bounds for dynamic pricing under semi-parametric forms (e.g. Luo et al., 2022; Xu & Wang, 2022; Fan et al., 2024) or additional shape constraints beyond Lipschitz continuity (e.g. Chen & Gallego, 2020), with detailed comparisons in Appendix A.

We conclude with a minimax lower bound for the scenario where only target data is available.

**Corollary 4.5.** *Let $\mathcal{I}(C_{\text{Lip}}, c_Q)$ denote the class of nonparametric dynamic pricing problems such that target data*

satisfy (1), Assumptions 2.1 and 2.2. It holds that

$$\inf_\pi \sup_{I \in \mathcal{I}(C_{\text{Lip}}, c_Q)} R_{\pi, I}(n_Q) \geq c n_Q^{\frac{d+2}{d+3}},$$

*where $c > 0$ is a constant only depending on constants $C_{\text{Lip}}$ and $c_Q$.*

It is worth noting that even without considering transfer learning (i.e. when only target data are available), the minimax lower bound in Corollary 4.5 is, to the best of our knowledge, the first result established under the Lipschitz assumption stated in Assumption 2.1. Previous works have only established minimax lower bounds either for the case with a semi-parametric form of the expected revenue function (e.g. Luo et al., 2022; Xu & Wang, 2022) or with additional assumption beyond Lipschitz condition on the reward function (e.g. Chen & Gallego, 2020).

## 5. Numerical experiments

In this section, we conduct numerical experiments to support our theoretical findings. Synthetic and real data analysis are in Sections 5.1 and 5.2, respectively. The code and datasets are available online[1].

### 5.1. Simulation studies

We conduct simulation studies under the covariate shift model as stated in (3). The target covariates $\{X_t\}_{t=1}^{n_Q}$ are independently and identically distributed (i.i.d.) according to $Q_X$, a uniform distribution over $[0, 1]^d$. For the source domain, covariate-price pairs $\{X_t^P, p_t^P\}_{t=1}^{n_P}$ are generated i.i.d. from a joint distribution $\mu$. We let the source marginal covariate distribution $P_X$ be chosen such that the density function $p_X(x)$ obeys $p_X(x) = c\|x - x^*\|_\infty^\gamma$, with $x^* = (1/2, \ldots, 1/2) \in \mathbb{R}^d$ and a normalization constant $c = 2^\gamma(\gamma + d)/d$. This choice of $p_X(x)$ ensures that $P_X$ satisfies Definition 2.3. The generation of the source prices is detailed in each scenario below. We consider the source data size $n_P \in \{10000(k-1) \colon k \in [5]\}$, the target data size $n_Q \in \{10000k \colon k \in [5]\}$ and the transfer exponent $\gamma \in \{0.5k \colon k \in [5]\}$.

To assess the performance of TLDP (Algorithm 1) in reducing target-domain regret, we compare it with TLDP utilising the target data only (Target-Only TLDP), the Adaptive Binning and Exploration (ABE) algorithm proposed in Chen & Gallego (2020) and the Explore-then-UCB (ExUCB) algorithm developed in Luo et al. (2022).

Through this subsection, TLDP refers to Algorithm 1 with $n_P = 2n_Q$. Note that both TLDP and Target-Only TLDP require the smallest exploration radius $\tilde{r}$ as an input and the

---

[1] https://github.com/chrisfanwang/dynamic-pricing

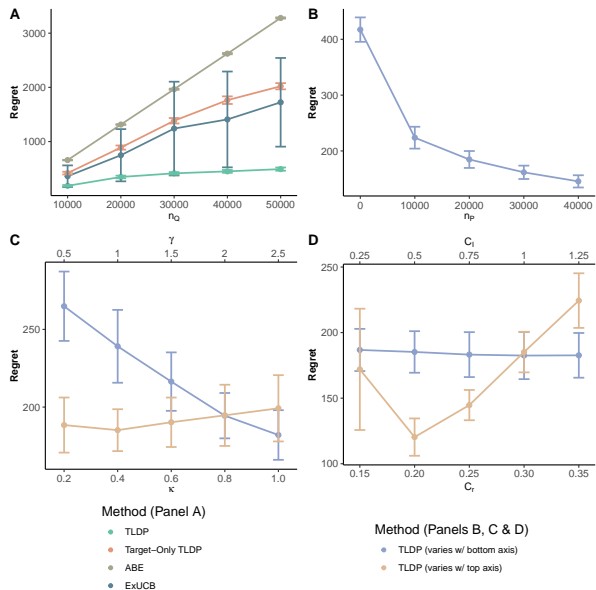

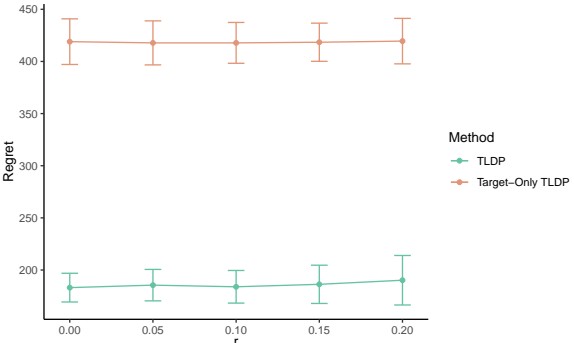

*Figure 2.* Results for Scenario 3 under posterior drift. The drift level $r$ varies, with the target data size $n_Q = 10000$, the source data size $n_P = 2n_Q$, the transfer exponent $\gamma = 1$, the exploration coefficient $\kappa = 1$, the index constant $C_I = 1$ and the exploration radius constant $C_r = 1/4$.

*Figure 1.* Results for Configuration 1 in Scenario 1. Panel (A) and (B): varying source data size $n_P$ and target data size $n_Q$, respectively. Panel (C) varying the transfer exponent $\gamma$ (top axis) and the exploration coefficient $\kappa$ (bottom axis). Panel (D): varying the index constant $C_I$ (top axis) and the exploration radius constant $C_r$ (bottom axis). For Panels (B), (C) and (D), we fix $n_Q = 10000$.

constant $C_I$ to construct the index defined in (9). Specifically, we set $C_I = 1$ and compute $\tilde{r}$ from (13) using the true values of the exploration coefficient $\kappa$, the transfer exponent $\gamma$ and $C_r = 1/4$. To evaluate robustness, we also conduct simulations using a list of mis-specified values of $\kappa$ and $\gamma$ when computing $\tilde{r}$. The corresponding simulation results can be found in Appendix D.1. Furthermore, to examine the sensitivity of TLDP to the choice of the constants, additional simulation studies are conducted for varying $C_I$ and $C_r$. For the ABE algorithm, we set $M = 0.1$, the constant used to define the maximal number observed in a level-$k$ bin in the partition, as suggested by Chen & Gallego (2020). For ExUCB, parameter choices include the phase exponent $\beta = 2/3$, the UCB exponent $\gamma = 1/6$, the exploration phase constant $C_1 = 1$, the discretization constant $C_2 = 20$ and the regularization parameter in UCB $\lambda = 0.1$, as suggested by Luo et al. (2022). In addition, we fix the maximum price $p_{\max} = 1$ and the maximum possible revenue $B = 1$ for the ExUCB implementation.

**Scenario 1.** The conditional density of the source price, given the source covariates, is defined as follows for any $x \in [0,1]^d$ and $p \in [0,1]$, $h_P(p|x) = \kappa$ if $p \in [p^* - r^*/2, p^* + r^*/2]$ and $h_P(p|x) = \{1 - \kappa r^*\}/(1 - r^*)$ otherwise, with $p^* = 1/2$, $r^* = 1/4$ and $\kappa \in \{0.2k : k \in [5]\}$. Furthermore, conditioned on the covariate $x$ and price $p$,

the observed rewards for both target and source datasets are independently and uniformly distributed in the interval $[f(x,p) - \nu, f(x,p) + \nu]$, where $f(\cdot, \cdot)$ denotes the expected reward function and $\nu = 0.1$. The expected reward function $f$ is constructed as $f(x,p) = \theta_0 + \theta^\top x p + \tilde{\theta} p^2$, for any $x \in [0,1]^d$, $p \in [0,1]$, where $\theta_0, \tilde{\theta} \in \mathbb{R}$ and $\theta \in \mathbb{R}^d$ are specified for each configuration.

We further consider two configurations. **Configuration 1.** Let the covariate space dimension be $d = 2$. The parameters of the reward function are set as $\theta_0 = 0.2$, $\theta = (0.3, 0.2)^\top$ and $\tilde{\theta} = -0.1$. **Configuration 2.** Let the covariate space dimension be $d = 3$. The parameters of the reward function are set as $\theta_0 = 0.3$, $\theta = (0.1, 0.4, -0.2)^\top$ and $\tilde{\theta} = -0.1$.

**Scenario 2.** The conditional density of the source price, given the source covariates, is defined as follows for any $x \in [0,1]^d$ and $p \in [0,1]$: $h_P(p|x) = \{1 - \kappa(1 - r^*)\}/r^*$ if $p \in [p^* - r^*/2, p^* + r^*/2]$ and $h_P(p|x) = \kappa$ otherwise, with $p^* = 1/2$, $r^* = 1/4$ and $\kappa \in \{0.2k : k \in [5]\}$. Further, conditioned on the covariate $x$ and price $p$, the observed rewards for both target and source datasets are independently distributed $Y \sim \mathcal{N}(f(x,p), \sigma^2)$ with $\sigma = 0.01$. For the expected reward function $f$, we adopt the setting from Cai et al. (2024a). Define a function $\phi \colon \mathbb{R}_+ \to [0,1]$ by $\phi(z) = 1$ for $0 \leq z < 1/12$, $\phi(z) = 2 - 12z$ for $1/12 \leq z < 1/6$ and $0 \phi(z) = 0$ otherwise. The reward function $f \colon [0,1]^d \times [0,1] \to [0,1]$ is then set to be $f(x,p) = 1/4 + \sum_{i=1}^m 3/4\phi\big(\|(x^\top, p)^\top - ((x_i^*)^\top, p_i^*)^\top\|_\infty\big) \mathbb{1}\{x \in B_{\mathcal{X}}(x_i^*, r^*)\}$, where $m > 0$, $r^* \in (0,1]$ and $\{((x_i^*)^\top, p_i^*)^\top\}_{i=1}^m$ are specified for each configuration.

We further consider two configurations. **Configuration 1.** Let the covariate space dimension be $d = 2$. Let $m = 4$, $r^* = 1/4$ and centres be $\{((x_i^*)^\top, p_i^*)^\top\}_{i=1}^m =$

$\{(1/4, 1/4, 1/4), (3/4, 1/4, 1/4), (1/4, 3/4, 3/4), (3/4, 3/4, 1/4)\}$. **Configuration 2.** Let the covariate space dimension be $d = 3$. Let $m = 8$, $r^* = 1/4$ and centres be $\{((x_i^*)^\top, p_i^*)^\top\}_{i=1}^m = \{(1/4, 1/4, 1/4, 1/4), (3/4, 1/4, 1/4, 1/4), (1/4, 3/4, 1/4, 1/4), (3/4, 3/4, 1/4, 1/4), (1/4, 1/4, 3/4, 3/4), (3/4, 1/4, 3/4, 3/4), (1/4, 3/4, 3/4, 3/4), (3/4, 3/4, 3/4, 3/4)\}$.

**Scenario 3.** Building on the setup of **Configuration 1** in **Scenario 1**, we fix the target reward parameters as $\theta_0 = 0.2$, $\theta = (0.3, 0.2)^\top$ and $\tilde{\theta} = -0.1$. We then modify the true source reward function to $f(x, p) = \theta_0' + x^\top \theta' p + \tilde{\theta}' p^2$, where $\theta_0' \sim \mathcal{N}(\theta_0, \sigma^2)$, $\theta' \sim \mathcal{N}(\theta, \sigma^2 I_d)$ and $\tilde{\theta}' \sim \mathcal{N}(\tilde{\theta}, \sigma^2)$. We set $\sigma = r \cdot \min\{|\theta_0|, \|\theta\|_\infty, |\tilde{\theta}|\}$, where the drift severity parameter $r \in \{0.05(k - 1) : k \in [5]\}$.

In **Scenarios 1** and **2**, we consider the covariate shift setting, where the reward functions are identical between the source and target domains. Specifically, in **Scenario 1**, the reward function is linear in the covariates and quadratic in the price, while in **Scenario 2**, it is fully nonparametric. In contrast, **Scenario 3** considers a posterior drift, where the reward functions differ between source and target domains, allowing us to assess the robustness of TLDP under model misspecification. The simulation results for **Configuration 1** of **Scenario 1** are presented in Figure 1 while results for **Scenario 3** are shown in Figure 2, Additional results are provided in Appendix D. Some key observations are highlighted in order.

In **Scenarios 1** and **2**, TLDP outperforms the alternatives. In **Scenario 1**, ExUCB outperforms ABE and Target-Only TLDP due to the linear structure of the reward function in the covariates. The ExUCB algorithm is specifically designed to leverage such linear relationships, thereby achieving superior performance. In **Scenario 2**, ABE and Target-Only TLDP demonstrate better performance than ExUCB. These findings align with the discussion in Section 4 and Appendix A.

In Panel (A), as the target data size $n_Q$ increases, the regret for all four methods increases. TLDP, however, exhibits a significantly slower rate of regret increase compared to its competitors, due to its ability to leverage information from the additional source data. Moreover, both TLDP and Target-Only TLDP demonstrate sublinear growth in regret as $n_Q$ increases, aligning with the theoretical regret order established in Theorem 4.1. In Panel (B), as the source data size $n_P$ increases, the regret of TLDP decreases, reflecting improved performance with larger source datasets. This trend aligns with the theoretical results in Theorem 4.1, where the regret is expected to decrease at the rate $n_P^{-1/(d+3+\gamma)}$, further validating the algorithm's performance.

In Panel (C), as the transfer exponent $\gamma$ increases, reflecting reduced overlap between source and target covariate distri-

*Table 1.* Results for the auto loan dataset with East South Central data as the target division. Columns correspond to the source divisions utilized in TLDP. Here, $n_P$ represents the number of the source data utilized in TLDP and $n$ denotes the total number of source observations. Each cell reports the mean and standard deviation over 100 simulations.

| METHODS | MOUNTAIN | PACIFIC |
|---|---|---|
| ABE | 70.81 (1.40) | 70.81 (1.40) |
| ExUCB | 64.91 (5.31) | 64.91 (5.31) |
| TLDP($n_P = 0$) | 71.95 (2.87) | 71.95 (2.87) |
| TLDP($n_P = 0.25n$) | 55.63 (8.04) | 51.61 (8.89) |
| TLDP($n_P = 0.5n$) | 54.29 (7.06) | 51.44 (7.12) |
| TLDP($n_P = 0.75n$) | 51.38 (6.92) | 49.40 (7.40) |
| TLDP($n_P = n$) | 50.23 (7.34) | 48.72 (8.27) |

butions, the regret of TLDP also increases. This indicates less efficient information transfer from the source to the target domain. Moreover, as the exploration coefficient $\kappa$ increases, the source data provide better exploration of the price space for each covariate. Consequently, the regret of TLDP decreases, indicating more effective information transfer. Panel (C) is consistent with Theorem 4.1, demonstrating the impact of these parameters on the efficiency of the transfer learning framework. It is evident from Panel (D) that the performance of TLDP remains relatively stable across different values of the index constant $C_I$ and the exploration radius constant $C_r$, indicating that TLDP is robust to the choice of the two constants.

In **Scenario 3**, Figure 2 demonstrates that TLDP continues to perform well under moderate posterior drift, indicating robustness beyond the covariate shift setting.

### 5.2. Real data analysis

In this section, we evaluate the practical utility of our proposed algorithm using the auto loan dataset (Phillips et al., 2015), which consists of $208,085$ applications submitted to a major online lender in the United States between July 2002 and November 2004. This dataset has been extensively studied in previous works, such as Phillips et al. (2015), Luo et al. (2024) and Zhao et al. (2024), to assess various dynamic pricing algorithms. The performance of TLDP is still compared against ABE and ExUCB.

The loan price is calculated as the net present value of future payments adjusted by the loan amount, as follows: Price = Monthly Payment $\times \sum_{i=1}^{\text{Term}} (1 + \text{Rate})^{-i} - \text{Loan Amount}$, where Rate $= 0.12\%$ represents the average monthly London interbank offered rate during the study period. The reward metric is defined as the product of the consumer's decision (whether the loan is accepted) and the computed price, capturing the total revenue generated for the lender.

For this analysis, five covariates identified as significant in prior studies (e.g. Luo et al., 2024; Zhao et al., 2024) are included: the loan amount approved, the approved term, the prime rate, the competitor's rate and the customer's FICO score. All features, including rewards, prices and covariates, are normalized to the range $[0, 1]$. Additionally, U.S. states are grouped into nine divisions following the United States Census Bureau classification: East North Central, East South Central, Middle Atlantic, Mountain, New England, Pacific, South Atlantic, West North Central and West South Central. Table 2 in Appendix D provides a sample of the processed dataset.

For this study, the East South Central division (8,062 applications) is designated as the target domain due to its smallest data size. The Mountain (12,527 applications), East North Central (20,686 applications), West South Central (25,695 applications) and Pacific (34,870 applications) divisions are used as source domains for the TLDP algorithm. The true reward function is approximated using a random forest model implemented in R (R Core Team, 2021), leveraging the `randomForest` package (Liaw & Wiener, 2002), trained on the target data. Optimal prices for each observation are determined by numerically optimizing the approximated reward function.

The results are averaged over 100 simulations, with each simulation randomly selecting 90% of the target data as the test data. All results can be found in Appendix D.2. Table 1 presents selected results with the key findings as follows. When no source data are utilized ($n_p = 0$), TLDP incurs slightly higher regret compared to ABE and ExUCB. While ExUCB achieves the lowest average cumulative regret, it exhibits significantly higher variance. Conversely, ABE demonstrates low variance but relatively high regret. As the source data size ($n_P$) increases, TLDP's cumulative regret consistently decreases, indicating its ability to effectively leverage data from other divisions.

## 6. Conclusion

In this paper, we study transfer learning for nonparametric contextual dynamic pricing under covariate shift, which, to the best of our knowledge, is the first time seen in the literature. We propose the TLDP algorithm, which adaptively partitions the covariate-price space and leverages source data information to guide pricing for the target data. We show that TLDP achieves optimal regret by establishing a matching minimax lower bound.

Our work offers several interesting directions for future research. First, the optimality of TLDP depends on prior knowledge of the transfer exponent $\gamma$ and the exploration coefficient $\kappa$, which are often unknown in practice. A natural extension is to estimate $\kappa$ from the source data before ana-

lyzing the target data and to adaptively update an estimate of $\gamma$ by measuring empirical overlaps between the source and target covariate distributions. Second, in many applications, multiple source datasets ($K > 1$) may be available, each with distinct transfer exponents $\gamma_k$ and exploration coefficients $\kappa_k$. A straightforward approach is to combine all sources into a single dataset before running TLDP. A more refined approach is to weight sources adaptively by estimating $\kappa_k$ and $\gamma_k$. Establishing regret bounds for such methods would be more challenging. Lastly, it is intriguing to explore transfer learning for nonparametric contextual dynamic pricing under posterior drift. In this framework, we need to quantify the discrepancies between the reward functions in the source and target domains and to develop novel methods that effectively leverage transferable information.

## Impact Statement

This paper presents work whose goal is to advance the field of Machine Learning. There are many potential societal consequences of our work, none which we feel must be specifically highlighted here.

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

# Appendix

All technical details of this paper can be found in the Appendix. A detailed comparison of our regret bound in Theorem 4.1 with those from non-transfer learning studies is presented in Appendix A. The proofs of Theorem 4.1 and Theorem 4.4 are included in Appendices B and C, respectively. Additional details and results for Section 5 are collected in Appendix D.

## A. Comparisons with non-transfer learning regret bounds

We compare the regret bound established for TLDP in Theorem 4.1 with those from relevant studies on dynamic pricing and contextual bandits that do not incorporate transfer learning:

- Slivkins (2011) studied contextual bandits with similarity information, where both contexts and actions are embedded in a metric space equipped with a distance function. The regret bounds in Slivkins (2011), adapted to our notation, are of order

$$n_Q^{\frac{d+2}{d+3}} \log(n_Q),$$

matching our result up to a logarithmic factor;

- Chen & Gallego (2020) investigated nonparametric dynamic pricing with covariates and proposed the ABE algorithm, which adaptively partitions the covariate space based on observed data to balance exploration and exploitation. They showed that the ABE algorithm achieves a regret bound of order

$$n_Q^{\frac{d+2}{d+4}} \log^2(n_Q),$$

which grows more slowly than ours due to additional assumptions beyond the standard Lipschitz condition stated in Assumption 2.1, such as the local strong concavity of the reward function;

- Several studies on nonparametric dynamic pricing (e.g., Luo et al., 2022; Xu & Wang, 2022; Fan et al., 2024) consider a specific form of the expected revenue function,

$$f(x,p) = p\{1 - F(p - x^\top \theta)\},$$

where $F$ is a nonparametric cumulative distribution function (CDF) of noise influencing customer valuations and $\theta \in \mathbb{R}^d$ is an unknown parameter vector representing customer sensitivity. For instance, Luo et al. (2022) developed the ExUCB algorithm and derived a regret bound of order

$$n_Q^{3/4}.$$

independent of $d$ in the exponent. This independence arises from the linear parametric form of $x^\top \theta$, which reduces the intrinsic complexity of the problem and mitigates the curse of dimensionality often encountered in fully nonparametric settings. In contrast, our results address a more general nonparametric reward function, which introduces additional complexity in capturing the covariate-price relationship without structural simplifications. Consequently, this leads to dimensional dependence, reflected in the $d$ appearing in the exponent of our regret bound;

- Bu et al. (2022) studied dynamic pricing under two partially linear demand models:

$$f(x,p) = p\{bp + g(x)\} \quad \text{and} \quad f(x,p) = p\{g'(p) + x^\top a\},$$

where $b < 0$, $a \in \mathbb{R}^d$, and $g, g'$ are unknown functions. They derived minimax regret bounds for both models. For the first model, assuming that $g$ is $\beta$-Hölder continuous, they established a minimax regret bound of order

$$\sqrt{T} \vee T^{\frac{d}{d+2\beta}}.$$

In comparison, our setting considers a reward function that is nonparametric in both the covariates and the price, leading to a regret bound with $d + 1$ in the exponent instead of $d$, thereby capturing the additional complexity introduced by the price dimension. For the second model, assuming that $g'$ is $k$th-order smooth with a smoothness parameter $\delta$, the minimax regret is of order

$$\sqrt{T} \vee (\delta T^{k+1})^{\frac{1}{2k+1}},$$

which is independent of $d$ in the exponent, as the model assumes a linear parametric structure in the covariates.

# B. Proof of Theorem 4.1

The proof of Theorem 4.1 is in Appendix B.1 with all necessary auxiliary results in Appendix B.2.

## B.1. Proof of Theorem 4.1

*Proof.* For any $t \leq n_Q$, let $\mathcal{A}_t$ denote the set of active balls at the beginning of time $t$. For any $B \in \mathcal{A}_t$, let $\mathbf{re}_t(B)$ and $n_t(B)$ denote the cumulative revenue and the cumulative count of $B$ from the target data before time $t$ and the source data $\mathcal{D}^P$. Further, define

$$\mathcal{E}_1^t = \left\{ \forall B \in \mathcal{A}_t \colon \left| v_t(B) - f(B) \right| \leq C_{\mathrm{Lip}} r(B) + \mathbf{conf}_t(B) \right\},$$

and

$$\mathcal{E}_2^t = \left\{ \forall B \in \mathcal{A}_t \colon \mathcal{E}_B \text{ holds} \right\} \quad \text{with} \quad \mathcal{E}_B = \left\{ n_B^P(\mathcal{D}^P) \geq C_{\mathcal{E}} \kappa n_P r(B)^{d+\gamma+1} \right\},$$

where $C_{\mathcal{E}} = c_\gamma c_Q$ and $f(B) = f(c(B))$. Intuitively speaking, $\mathcal{E}_1^t$ represents the good event where the empirical average revenue $v_t(B)$ for every active ball $B \in \mathcal{A}_t$ lies within the upper confidence bound around the true expected revenue $f(B)$, while $\mathcal{E}_2^t$ states that the source data (measured by the number) can still provide sufficient information up to time $t$.

**Consequences of $\mathcal{E}_1^t$:** Suppose for any time point $t$, $\mathcal{E}_1^t$ holds. Then by (10), the condition $C_I \geq C_{\mathrm{Lip}}$ and the event $\mathcal{E}_1^t$, for any $B \in \mathcal{A}_t$, we have

$$I_t^{\mathbf{pre}}(B) \geq f(B). \tag{15}$$

For any $x \in \mathcal{X}$, denote $p^*(x) = \min\{p' \in \arg\max_{p \in \mathcal{P}} f(x, p)\}$. By Lemma B.3, there exists a $B \in \mathcal{A}_t$ such that $(X_t, p^*(X_t)) \in \mathbf{dom}(B, \mathcal{A}_t)$. Let $B_t^{\mathbf{sel}}$ denote the selected ball at time $t$, then we derive that for such $B$,

$$\begin{aligned}
I_t(B_t^{\mathbf{sel}}) \geq & I_t(B) \\
= & C_I r(B) + \min_{B' \in \mathcal{A}_t} \left\{ I_t^{\mathbf{pre}}(B') + C_I \|c(B) - c(B')\|_\infty \right\} \\
\geq & C_I r(B) + \min_{B' \in \mathcal{A}_t} \left\{ f(B') + C_I \|c(B) - c(B')\|_\infty \right\} \\
\geq & C_I r(B) + f(B) \geq f^*(X_t),
\end{aligned} \tag{16}$$

where the first inequality follows from the selection rule in Algorithm 1 (Step 4 therein), the first equality follows from (9), the second inequality follows from (15), and the third and the last inequalities follow from Assumption 2.1 and $C_I \geq C_{\mathrm{Lip}}$.

Let $B_t^{\mathbf{par}}$ be the parent of $B_t^{\mathbf{sel}}$, we have that (due to step 9 in Algorithm 1)

$$\|c(B_t^{\mathbf{par}}) - c(B_t^{\mathbf{sel}})\|_\infty \leq r(B_t^{\mathbf{par}}). \tag{17}$$

Further note that $B_t^{\mathbf{par}}$ must satisfy the condition in Step 7 in Algorithm 1 (in order to produce $B_t^{\mathbf{sel}}$), we have $n_t(B_t^{\mathbf{par}}) - n_{B_t^{\mathbf{par}}}^P(\mathcal{D}^P) \geq T_{B_t^{\mathbf{par}}}^Q(\mathcal{D}^P)$. Then

$$\begin{aligned}
\mathbf{conf}_t(B_t^{\mathbf{par}}) = & 2 \sqrt{\frac{\log \left\{ n_Q \vee (\kappa n_p)^{\frac{d+3}{d+3+\gamma}} \right\}}{n_t(B_t^{\mathbf{par}})}} \leq 2 \sqrt{\frac{\log \left\{ n_Q \vee (\kappa n_p)^{\frac{d+3}{d+3+\gamma}} \right\}}{T_{B_t^{\mathbf{par}}}^Q + n_{B_t^{\mathbf{par}}}^P(\mathcal{D}^P)}} \\
\leq & 2 \sqrt{\frac{\log \left\{ n_Q \vee (\kappa n_p)^{\frac{d+3}{d+3+\gamma}} \right\}}{\omega(B_t^{\mathbf{par}})}} \leq 2 r(B_t^{\mathbf{par}}),
\end{aligned} \tag{18}$$

where the second inequality follows from (11) and the last form (12). Note that

$$\begin{aligned}
I_t^{\mathbf{pre}}(B_t^{\mathbf{par}}) = & v_t(B_t^{\mathbf{par}}) + C_I r(B_t^{\mathbf{par}}) + \mathbf{conf}_t(B_t^{\mathbf{par}}) \\
\leq & f(B_t^{\mathbf{par}}) + (C_I + C_{\mathrm{Lip}}) r(B_t^{\mathbf{par}}) + 2 \mathbf{conf}_t(B_t^{\mathbf{par}}) \\
\leq & f(B_t^{\mathbf{par}}) + (4 + C_I + C_{\mathrm{Lip}}) r(B_t^{\mathbf{par}}) \leq f(B_t^{\mathbf{sel}}) + (4 + C_I + 2 C_{\mathrm{Lip}}) r(B_t^{\mathbf{par}}),
\end{aligned} \tag{19}$$

where the first equality follows from (10), the first inequality follows from the event $\mathcal{E}_1^t$, the second inequality follows form (18) and the last inequality follows from Assumption 2.1 and (17). Note that

$$
\begin{aligned}
I_t\big(B_t^{\mathbf{sel}}\big) \leq & C_I r\big(B_t^{\mathbf{sel}}\big) + I_t^{\mathbf{pre}}\big(B_t^{\mathbf{par}}\big) + C_I \|c(B_t^{\mathbf{sel}}) - c(B_t^{\mathbf{par}})\|_\infty \\
\leq & C_I r\big(B_t^{\mathbf{sel}}\big) + I_t^{\mathbf{pre}}\big(B_t^{\mathbf{par}}\big) + C_I r\big(B_t^{\mathbf{par}}\big) \\
\leq & C_I r\big(B_t^{\mathbf{sel}}\big) + f\big(B_t^{\mathbf{sel}}\big) + (4 + 2C_I + 2C_{\mathrm{Lip}}) r\big(B_t^{\mathbf{par}}\big) \\
\leq & f\big(B_t^{\mathbf{sel}}\big) + (8 + 5C_I + 4C_{\mathrm{Lip}}) r\big(B_t^{\mathbf{sel}}\big) \leq f\big(X_t, p_t\big) + (8 + 5C_I + 5C_{\mathrm{Lip}}) r\big(B_t^{\mathbf{sel}}\big),
\end{aligned}
\tag{20}
$$

where the first inequality holds due to (9), second inequality follows from (17), the third inequality follows from (19), the forth holds due to $r\big(B_t^{\mathbf{sel}}\big) = r\big(B_t^{\mathbf{par}}\big)/2$, and the last inequality follows from Assumption 2.1 and that $(X_t, p_t) \in B_t^{\mathbf{sel}}$.

Combining (16) and (20), we have that

$$
f^*(X_t) - f\big(X_t, p_t\big) \leq C_1 r\big(B_t^{\mathbf{sel}}\big),
\tag{21}
$$

where $C_1 = (8 + 5C_I + 5C_{\mathrm{Lip}})$.

**Step 1: Consider the case where $n_Q \geq (\kappa n_P)^{\frac{d+3}{d+3+\gamma}}$.**

**Regret decomposition**: We then decompose the regret into several terms

$$
\begin{aligned}
R_\pi(n_Q) = & \mathbb{E}\bigg\{ \sum_{t=1}^{n_Q} \big\{ f^*(X_t) - f(X_t, p_t) \big\} \bigg\} \\
= & \mathbb{E}\bigg[ \sum_{t=1}^{n_Q} \big\{ f^*(X_t) - f(X_t, p_t) \big\} \mathbb{1}_{\{(\mathcal{E}_1^t)^c\}} \bigg] + \mathbb{E}\bigg[ \sum_{t=1}^{n_Q} \big\{ f^*(X_t) - f(X_t, p_t) \big\} \mathbb{1}_{\{\mathcal{E}_1^t\}} \bigg] \\
:= & (\mathrm{I}) + (\mathrm{II}).
\end{aligned}
\tag{22}
$$

In the following, we deal with the above two terms separately.

**Step 1.1: Bound for (I).** By a union bound argument and Lemma B.1, we have that

$$
\mathbb{P}\big\{(\mathcal{E}_1^t)^c\big\} \leq \sum_{B \in \mathcal{A}_t} 18 n_Q^{-2} \tilde{r}^{-2} \log(n_Q) \leq 18 t n_Q^{-2} \tilde{r}^{-3} \log(n_Q) \leq 18 n_Q^{-1} \tilde{r}^{-3} \log(n_Q),
\tag{23}
$$

where the second inequality holds by noting $|\mathcal{A}_t| \leq t\tilde{r}^{-1}$ since the smallest radius is lower bounded by $\tilde{r}$ and hence at most $\tilde{r}^{-1}$ balls can be activated at each time point; and the third inequality holds by $t \leq n_Q$.

Therefore, recall that $0 \leq f(X_t, p_t) \leq f^*(X_t) \leq 1$, it holds that

$$
(\mathrm{I}) \leq \sum_{t=1}^{n_Q} \mathbb{P}\{\mathcal{E}_1^c\} \leq 18 \tilde{r}^{-3} \log(n_Q) \leq 4C_1 \tilde{r}^{-3} \log(n_Q).
\tag{24}
$$

**Step 1.2: Bound for (II).** For any $r \in (0, 1]$, let $\mathcal{F}_r = \{B \in \mathcal{A}_{n_Q} : r(B) = r\}$. For any $B \subset \mathcal{Z}$, let $\mathcal{S}^Q(B)$ be the set of times $s \in [n_Q]$ when ball $B$ was selected for the target data. By Lemma B.3, we have that

$$
|\mathcal{F}_r| \leq N_r^{\mathrm{Pack}}(\mathcal{Z}) \leq N_{r/2}(\mathcal{Z}) \leq \left(\frac{2}{r}\right)^{d+1},
\tag{25}
$$

where $N_r^{\mathrm{Pack}}(\mathcal{Z})$ and $N_r(\mathcal{Z})$ denote $r$-packing number and $r$-covering number of $\mathcal{Z}$, respectively, and the second inequality follows from the fact that $N_{2r}^{\mathrm{Pack}}(\mathcal{Z}) \leq N_r(\mathcal{Z})$.

Then, note that

$$
\begin{aligned}
\text{(II)} =& \mathbb{E}\left[\sum_{t=1}^{n_Q}\left\{f^*(X_t)-f(X_t,p_t)\right\}\mathbb{1}_{\{\mathcal{E}_1^t\}}\right]\\
\leq& \mathbb{E}\left[\sum_{r:\tilde{r}\leq r\leq 1}\sum_{B\in\mathcal{F}_r}\sum_{t\in\mathcal{S}^Q(B)}\left\{f^*(X_t)-f(X_t,p_t)\right\}\mathbb{1}_{\{\mathcal{E}_1^t\}}\right]\\
=& \mathbb{E}\left[\sum_{B\in\mathcal{F}_{\tilde{r}}}\sum_{t\in\mathcal{S}^Q(B)}\left\{f^*(X_t)-f(X_t,p_t)\right\}\mathbb{1}_{\{\mathcal{E}_1^t\}}\right]\\
&+ \mathbb{E}\left[\sum_{r:\tilde{r}<r\leq 1}\sum_{B\in\mathcal{F}_r}\sum_{t\in\mathcal{S}^Q(B)}\left\{f^*(X_t)-f(X_t,p_t)\right\}\mathbb{1}_{\{\mathcal{E}_1^t\}}\right]\\
\leq& C_1 n_Q\tilde{r} + \mathbb{E}\left\{\sum_{r:\tilde{r}<r\leq 1}\sum_{B\in\mathcal{F}_r}\sum_{t\in\mathcal{S}^Q(B)}C_1 r\mathbb{1}_{\{\mathcal{E}_1^t\}}\right\},
\end{aligned}
\tag{26}
$$

where the second inequality holds due to (21).

When $n_Q \geq (\kappa n_P)^{\frac{d+3}{d+3+\gamma}}$, by (11) and (12), for any $r(B) > \tilde{r}$, we have

$$
|\mathcal{S}^Q(B)| \leq T_B^Q \leq \omega(B) \leq \frac{\log(n_Q)}{r(B)^2}+1.
\tag{27}
$$

as otherwise a new ball within $B$ will be activated due to Step 6 in Algorithm 1. As a consequence, by (25), (26) and (27), we have that

$$
\begin{aligned}
\text{(II)} \leq& C_1 n_Q\tilde{r} + \sum_{r:\tilde{r}<r\leq 1}C_1 r\left(\frac{2}{r}\right)^{d+1}\left\{\frac{\log(n_Q)}{r^2}+1\right\}\\
\leq& C_1 n_Q\tilde{r} + 2^{d+1}C_1\sum_{r:\tilde{r}<r\leq 1}\left\{r^{-(d+2)}\log(n_Q)+r^{-d}\right\}\\
\leq& C_1 n_Q\tilde{r} + 2^{d+2}C_1\left\{\tilde{r}^{-(d+2)}\log(n_Q)+\tilde{r}^{-d}\right\},
\end{aligned}
\tag{28}
$$

where the last inequality follows from the fact that

$$
\sum_{r:\tilde{r}<r\leq 1}r^{-d} = \sum_{k=0}^{\lceil -\log_2(\tilde{r})\rceil-1}2^{kd} = \frac{2^{\lceil -\log_2(\tilde{r})\rceil d}-1}{2^d-1} \leq \frac{2^d\tilde{r}^{-d}-1}{2^d-1} \leq \frac{2^d}{2^d-1}\tilde{r}^{-d} \leq 2\tilde{r}^{-d}.
$$

Combining (22), (24) and (28), we have that in the case where $n_Q \geq (\kappa n_P)^{\frac{d+3}{d+3+\gamma}}$,

$$
\begin{aligned}
R_\pi(n_Q) \leq& 4C_1\tilde{r}^{-3}\log(n_Q) + C_1 n_Q\tilde{r} + 2^{d+2}C_1\left\{\tilde{r}^{-(d+2)}\log(n_Q)+\tilde{r}^{-d}\right\}\\
\leq& C_1 n_Q\tilde{r} + 2^{d+4}C_1\tilde{r}^{-(d+2)}\log(n_Q).
\end{aligned}
\tag{29}
$$

**Step 2**: Consider the case where $n_Q < (\kappa n_P)^{\frac{d+3}{d+3+\gamma}}$ and

$$
\tilde{r} \geq \left\{\frac{8\log\left\{(\kappa n_p)^{\frac{d+3}{d+3+\gamma}}\right\}}{C_{\mathcal{E}}\kappa n_P}\right\}^{\frac{1}{d+3+\gamma}},
\tag{30}
$$

where $C_{\mathcal{E}} = c_\gamma c_Q$.

**Regret decomposition**: We then decompose the regret into several terms

$$
\begin{aligned}
R_\pi(n_Q) =& \mathbb{E}\bigg\{ \sum_{t=1}^{n_Q} \Big\{ f^*(X_t) - f(X_t, p_t) \Big\} \bigg\} \\
=& \mathbb{E}\bigg[ \sum_{t=1}^{n_Q} \Big\{ f^*(X_t) - f(X_t, p_t) \Big\} \mathbb{1}_{\{(\mathcal{E}_1^t)^c\}} \bigg] + \mathbb{E}\bigg[ \sum_{t=1}^{n_Q} \Big\{ f^*(X_t) - f(X_t, p_t) \Big\} \mathbb{1}_{\{\mathcal{E}_1^t \cap \mathcal{E}_2^t\}} \bigg] \\
& + \mathbb{E}\bigg[ \sum_{t=1}^{n_Q} \Big\{ f^*(X_t) - f(X_t, p_t) \Big\} \mathbb{1}_{\{\mathcal{E}_1^t \cap (\mathcal{E}_2^t)^c\}} \bigg] \\
:=& (\mathrm{I}) + (\mathrm{II}) + (\mathrm{III}).
\end{aligned}
\tag{31}
$$

In the following, we deal with the above three terms separately.

**Step 2.1: Bound for (I).** Similar to (23), we have that

$$
\mathbb{P}\big\{ (\mathcal{E}_1^t)^c \big\} \leq \sum_{B \in \mathcal{A}_t} 3\{ (\kappa n_P)^{\frac{d+3}{d+3+\gamma}} \}^{-2} \leq 3t\tilde{r}^{-1} \{ (\kappa n_P)^{\frac{d+3}{d+3+\gamma}} \}^{-2} \leq 3n_Q \tilde{r}^{-1} \{ (\kappa n_P)^{\frac{d+3}{d+3+\gamma}} \}^{-2}.
\tag{32}
$$

Therefore, recall that $0 \leq f(X_t, p_t) \leq f^*(X_t) \leq 1$, it holds that

$$
(\mathrm{I}) \leq \sum_{t=1}^{n_Q} \mathbb{P}\{\mathcal{E}_1^c\} \leq 3n_Q^2 \tilde{r}^{-1} \{ (\kappa n_P)^{\frac{d+3}{d+3+\gamma}} \}^{-2}.
\tag{33}
$$

**Step 2.2: Bound for (II).** By similar arguments as (26), we have

$$
(\mathrm{II}) \leq C_1 n_Q \tilde{r} + \mathbb{E}\bigg[ \sum_{r:\tilde{r} < r \leq 1} \sum_{B \in \mathcal{F}_r} \sum_{t \in \mathcal{S}^Q(B)} \big\{ f^*(X_t) - f(X_t, p_t) \big\} \mathbb{1}_{\{\mathcal{E}_1^t \cap \mathcal{E}_2^t\}} \bigg].
\tag{34}
$$

Note that under $n_Q < (\kappa n_P)^{\frac{d+3}{d+3+\gamma}}$ and (30), if $\mathcal{E}_2^t$ holds, by Lemma B.4, we have that $T_B^Q = 0$ for any $B \in \mathcal{A}_t$. This implies that $S^Q(B) = \emptyset$. Therefore, the second term in (34) vanishes and thus we have,

$$
(\mathrm{II}) \leq C_1 n_Q \tilde{r}.
\tag{35}
$$

**Step 2.3: Bound for (III).** Note that by Lemma B.4 and a union bound argument, under $n_Q < (\kappa n_P)^{\frac{d+3}{d+3+\gamma}}$ and (30), it holds that

$$
\mathbb{P}\big\{ (\mathcal{E}_2^t)^c \big\} \leq n_Q \tilde{r}^{-1} \{ (\kappa n_P)^{\frac{d+3}{d+3+\gamma}} \}^{-2}.
$$

Therefore,

$$
(\mathrm{III}) \leq \sum_{t=1}^{n_Q} \mathbb{P}\big\{ (\mathcal{E}_2^t)^c \big\} \leq n_Q^2 \tilde{r}^{-1} \{ (\kappa n_P)^{\frac{d+3}{d+3+\gamma}} \}^{-2}.
\tag{36}
$$

Combining (31), (33), (35) and (36), we can conclude that

$$
\begin{aligned}
R_\pi(n_Q) \leq& 3n_Q^2 \tilde{r}^{-1} \{ (\kappa n_P)^{\frac{d+3}{d+3+\gamma}} \}^{-2} + C_1 n_Q \tilde{r} + n_Q^2 \tilde{r}^{-1} \{ (\kappa n_P)^{\frac{d+3}{d+3+\gamma}} \}^{-2} \\
\leq& 4n_Q^2 \tilde{r}^{-1} \{ (\kappa n_P)^{\frac{d+3}{d+3+\gamma}} \}^{-2} + C_1 n_Q \tilde{r} \leq 2C_1 n_Q \tilde{r},
\end{aligned}
\tag{37}
$$

where the last inequality follows from (30).

**Step 3: Consider the case where $n_Q < (\kappa n_P)^{\frac{d+3}{d+3+\gamma}}$ and**

$$
\tilde{r} < \bigg\{ \frac{8 \log \big\{ (\kappa n_p)^{\frac{d+3}{d+3+\gamma}} \big\}}{C_{\mathcal{E}} \kappa n_P} \bigg\}^{\frac{1}{d+3+\gamma}}.
$$

**Regret decomposition**: We then decompose the regret into several terms

$$
\begin{aligned}
R_\pi(n_Q) =& \mathbb{E}\bigg\{\sum_{t=1}^{n_Q}\Big\{f^*(X_t) - f(X_t, p_t)\Big\}\bigg\} \\
=& \mathbb{E}\bigg[\sum_{t=1}^{n_Q}\Big\{f^*(X_t) - f(X_t, p_t)\Big\}\mathbb{1}_{\{(\mathcal{E}_1^t)^c\}}\bigg] + \mathbb{E}\bigg[\sum_{t=1}^{n_Q}\Big\{f^*(X_t) - f(X_t, p_t)\Big\}\mathbb{1}_{\{\mathcal{E}_1^t\}}\bigg] \\
:=& (\mathrm{I}) + (\mathrm{II}).
\end{aligned}
\tag{38}
$$

In the following, we deal with the above two terms separately.

**Step 3.1: Bound for (I).** By the same argument as (23), we can show that

$$
\mathbb{P}\big\{(\mathcal{E}_1^t)^c\big\} \leq 18 n_Q^{-1}\tilde{r}^{-3}\log\big\{(\kappa n_p)^{\frac{d+3}{d+3+\gamma}}\big\},
\tag{39}
$$

which implies that

$$
(\mathrm{I}) \leq \sum_{t=1}^{n_Q}\mathbb{P}\big\{(\mathcal{E}_1^t)^c\big\} \leq 18\tilde{r}^{-3}\log\big\{(\kappa n_p)^{\frac{d+3}{d+3+\gamma}}\big\}.
\tag{40}
$$

**Step 3.2: Bound for (II).** By similar arguments as (26), we have

$$
(\mathrm{II}) \leq C_1 n_Q\tilde{r} + \mathbb{E}\bigg\{\sum_{r:\tilde{r}<r\leq 1}\sum_{B\in\mathcal{F}_r}\sum_{t\in\mathcal{S}^Q(B)}C_1 r\mathbb{1}_{\{\mathcal{E}_1^t\}}\bigg\}.
\tag{41}
$$

When $n_Q < (\kappa n_P)^{\frac{d+3}{d+3+\gamma}}$, similar to (27), we have

$$
|\mathcal{S}^Q(B)| \leq T_B^Q \leq \omega(B) \leq \frac{\log\big\{(\kappa n_p)^{\frac{d+3}{d+3+\gamma}}\big\}}{r(B)^2} + 1.
\tag{42}
$$

As a consequence, by (25), (41) and (42), we have that

$$
\begin{aligned}
(\mathrm{II}) \leq& C_1 n_Q\tilde{r} + \sum_{r:\tilde{r}<r\leq 1}C_1 r\Big(\frac{2}{r}\Big)^{d+1}\bigg\{\frac{\log\big\{(\kappa n_p)^{\frac{d+3}{d+3+\gamma}}\big\}}{r^2} + 1\bigg\} \\
\leq& C_1 n_Q\tilde{r} + 2^{d+1}C_1\sum_{r:\tilde{r}<r\leq 1}\Big\{r^{-(d+2)}\log\big\{(\kappa n_p)^{\frac{d+3}{d+3+\gamma}}\big\} + r^{-d}\Big\} \\
\leq& C_1 n_Q\tilde{r} + 2^{d+2}C_1\Big\{\tilde{r}^{-(d+2)}\log\big\{(\kappa n_p)^{\frac{d+3}{d+3+\gamma}}\big\} + \tilde{r}^{-d}\Big\}.
\end{aligned}
\tag{43}
$$

Combining (38), (40) and (43), in the case where $n_Q < (\kappa n_P)^{\frac{d+3}{d+3+\gamma}}$ and

$$
\tilde{r} \leq \bigg\{\frac{8\log\big\{(\kappa n_p)^{\frac{d+3}{d+3+\gamma}}\big\}}{C_\mathcal{E}\kappa n_P}\bigg\}^{\frac{1}{d+3+\gamma}},
\tag{44}
$$

we have that

$$
\begin{aligned}
R_\pi(n_Q) \leq& 18\tilde{r}^{-3}\log\big\{(\kappa n_p)^{\frac{d+3}{d+3+\gamma}}\big\} + C_1 n_Q\tilde{r} + 2^{d+2}C_1\Big\{\tilde{r}^{-(d+2)}\log\big\{(\kappa n_p)^{\frac{d+3}{d+3+\gamma}}\big\} + \tilde{r}^{-d}\Big\} \\
\leq& 2^{d+4}C_1\tilde{r}^{-(d+2)}\log\big\{(\kappa n_p)^{\frac{d+3}{d+3+\gamma}}\big\},
\end{aligned}
\tag{45}
$$

where the last inequality follows from (44).

**Step 4: Combining all results.** Combining (29), (37) and (45), we have that

$$
R_\pi(n_Q) \leq
\begin{cases}
C_1 n_Q \tilde{r} + 2^{d+3} C_1 \tilde{r}^{-(d+2)} \log(n_Q), & \text{if } n_Q \geq (\kappa n_P)^{\frac{d+3}{d+3+\gamma}}, \\
2 C_1 n_Q \tilde{r}, & \text{if } n_Q < (\kappa n_P)^{\frac{d+3}{d+3+\gamma}} \text{ and } \tilde{r} \geq \left\{ \frac{8 \log \left\{ (\kappa n_p)^{\frac{d+3}{d+3+\gamma}} \right\}}{C_{\mathcal{E}} \kappa n_P} \right\}^{\frac{1}{d+3+\gamma}}, \\
2^{d+4} C_1 \tilde{r}^{-(d+2)} \log \left\{ (\kappa n_p)^{\frac{d+3}{d+3+\gamma}} \right\}, & \text{otherwise.}
\end{cases}
$$

By (13), $C_r^4 c_\gamma c_Q \geq 8$, we can conclude that

$$
R_\pi(n_Q) \leq C n_Q \left\{ n_Q + (\kappa n_P)^{\frac{d+3}{d+3+\gamma}} \right\}^{-\frac{1}{d+3}} \log^{\frac{1}{d+3}} \left\{ n_Q + (\kappa n_P)^{\frac{d+3}{d+3+\gamma}} \right\},
$$

where $C > 0$ is a constant only depending on constants $C_I$, $C_r$, $C_{\text{Lip}}$, $c_\gamma$ and $c_Q$. We complete the proof. $\qquad \square$

## B.2. Auxiliary results

**Lemma B.1.** *For Algorithm 1, at the beginning of the round $t$ for $t \leq n_Q$, if ball $B \in \mathcal{A}_t$, then under Assumptions 2.1 and 2.2 , we have that*

$$
\mathbb{P}\left\{ |v_t(B) - f(B)| \leq C_{\text{Lip}} r(B) + \boldsymbol{conf}_t(B) \right\} \geq 1 - \epsilon_{\tilde{r}},
$$

*where $C_{\text{Lip}} > 0$ is defined in Assumption 2.1 and*

$$
\epsilon_{\tilde{r}} =
\begin{cases}
18 n_Q^{-2} \tilde{r}^{-2} \log(n_Q), & \text{if } n_Q \geq (\kappa n_P)^{\frac{d+3}{d+3+\gamma}}, \\
3 \{ (\kappa n_P)^{\frac{d+3}{d+3+\gamma}} \}^{-2}, & \text{if } n_Q < (\kappa n_P)^{\frac{d+3}{d+3+\gamma}} \text{ and } \tilde{r} \geq \left\{ \frac{8 \log \left\{ (\kappa n_p)^{\frac{d+3}{d+3+\gamma}} \right\}}{C_{\mathcal{E}} \kappa n_P} \right\}^{\frac{1}{d+3+\gamma}}, \\
18 n_Q^{-2} \tilde{r}^{-2} \log \left\{ (\kappa n_p)^{\frac{d+3}{d+3+\gamma}} \right\}, & \text{otherwise.}
\end{cases}
\tag{46}
$$

*Here $C_{\mathcal{E}} = c_\gamma c_Q$ with constants $c_\gamma$ defined in Definition 2.3 and Assumption 2.2, respectively.*

*Proof.* Fix time $t$ and a ball $B \in \mathcal{A}_t$, and recall its centre as $c(B)$. Let $\mathcal{S}_t^Q(B)$ be the set of times $s \in [t-1]$ when ball $B$ was selected for the target data, with $|\mathcal{S}_t^Q(B)| = n_t^Q(B)$. Let $\mathcal{S}^P(B)$ be the set of times $s \in [n_P]$ when ball $B$ was fallen for the source data, with $|\mathcal{S}^P(B)| = n_B^P(\mathcal{D}^P)$. Note that $n_t(B) = n_t^Q(B) + n_B^P(\mathcal{D}^P)$ and

$$
v_t(B) = \frac{1}{n_t(B)} \left\{ \sum_{s \in \mathcal{S}_t^Q(B)} Y_s + \sum_{s \in \mathcal{S}^P(B)} Y_s^P \right\}.
$$

Denote

$$
\tilde{v}_t(B) = \frac{1}{n_t(B)} \left\{ \sum_{s \in \mathcal{S}_t^Q(B)} f(X_s, p_s) + \sum_{s \in \mathcal{S}^P(B)} f(X_s^P, p_s^P) \right\}.
$$

By Lemma B.2, it holds that

$$
\mathbb{P}\left\{ |v_t(B) - \tilde{v}_t(B)| > U\left( n_t^Q(B), n_B^P(\mathcal{D}^P), \delta_1, \delta_2 \right) \Big| \{ (X_t^P, p_t^P) \}_{t=1}^{n_P} \right\} \leq 8 \delta_1 T_B^Q + 2 \delta_2,
$$

where $T_B^Q$ is defined in (11) and

$$
U\left( n_t^Q(B), n_B^P(\mathcal{D}^P), \delta_1, \delta_2 \right) =
\begin{cases}
\sqrt{\frac{2 \log(1/\delta_1)}{n_t^Q(B) + n_B^P(\mathcal{D}^P)}} & \text{if } n_t^Q(B) > 0, \\
\sqrt{\frac{2 \log(1/\delta_2)}{n_B^P(\mathcal{D}^P)}} & \text{if } n_t^Q(B) = 0.
\end{cases}
$$

Let $\delta_1 = n_Q^{-2}$ and $\delta_2 = \{ n_Q \vee (\kappa n_P)^{\frac{d+3}{d+3+\gamma}} \}^{-2}$. By (8), we have that

$$
\mathbb{P}\left\{ |v_t(B) - \tilde{v}_t(B)| > \boldsymbol{conf}_t(B) \Big| \{ (X_t^P, p_t^P) \}_{t=1}^{n_P} \right\} \leq 8 n_Q^{-2} T_B^Q + 2 \{ n_Q \vee (\kappa n_P)^{\frac{d+3}{d+3+\gamma}} \}^{-2}.
\tag{47}
$$

Furthermore, note that

$$\left|\tilde{v}_t(B) - f(B)\right| \leq \frac{1}{n_t(B)} \left\{ \sum_{s \in \mathcal{S}_t^Q(B)} \left|f(X_s, p_s) - f(c(B))\right| + \sum_{s \in \mathcal{S}_P(B)} \left|f(X_s^P, p_s^P) - f(c(B))\right| \right\}$$

$$\leq C_{\text{Lip}} r(B), \tag{48}$$

where the second inequality follows form Assumption 2.1. Finally, combining (47) and (48), we have that

$$\mathbb{P}\left\{ \left|v_t(B) - f(B)\right| > C_{\text{Lip}} r(B) + \mathbf{conf}_t(B) \Big| \{(X_t^P, p_t^P)\}_{t=1}^{n_P} \right\}$$

$$\leq 8 n_Q^{-2} T_B^Q + 2\{ n_Q \vee (\kappa n_P)^{\frac{d+3}{d+3+\gamma}} \}^{-2}. \tag{49}$$

In what follows, we consider different cases in (46).

**Case 1.** In this case, we consider $n_Q \geq (\kappa n_P)^{\frac{d+3}{d+3+\gamma}}$. Then, we have that

$$8 n_Q^{-2} T_B^Q + 2\{ n_Q \vee (\kappa n_P)^{\frac{d+3}{d+3+\gamma}} \}^{-2} \leq 8 n_Q^{-2} \omega(B) + 2 n_Q^{-2} \leq 8 n_Q^{-2} r(B)^{-2} \log(n_Q) + 10 n_Q^{-2}$$

$$\leq 8 n_Q^{-2} \tilde{r}^{-2} \log(n_Q) + 10 n_Q^{-2}$$

$$\leq 18 n_Q^{-2} \tilde{r}^{-2} \log(n_Q), \tag{50}$$

where the first inequality follows from (11), the second inequality follows from (12), and the third inequality follows from $r(B) \geq \tilde{r}$. Then combining (49) and (50), we have that

$$\mathbb{P}\left\{ \left|v_t(B) - f(B)\right| > C_{\text{Lip}} r(B) + \mathbf{conf}_t(B) \right\}$$

$$= \mathbb{E}\left[ \mathbb{P}\left\{ \left|v_t(B) - f(B)\right| > C_{\text{Lip}} r(B) + \mathbf{conf}_t(B) \Big| \{(X_t^P, p_t^P)\}_{t=1}^{n_P} \right\} \right]$$

$$\leq 18 n_Q^{-2} \tilde{r}^{-2} \log(n_Q). \tag{51}$$

**Case 2.** In this case, we consider $n_Q < (\kappa n_P)^{\frac{d+3}{d+3+\gamma}}$, and

$$\tilde{r} \geq \left\{ \frac{8 \log \left\{ (\kappa n_p)^{\frac{d+3}{d+3+\gamma}} \right\}}{C_{\mathcal{E}} \kappa n_P} \right\}^{\frac{1}{d+3+\gamma}}.$$

Define the event

$$\mathcal{E}_B = \left\{ n_B^P(\mathcal{D}^P) \geq C_{\mathcal{E}} \kappa n_P r(B)^{d+\gamma+1} \right\}.$$

By Lemma B.4, we have that

$$\mathbb{P}\{\mathcal{E}_B^c\} \leq \{ (\kappa n_P)^{\frac{d+3}{d+3+\gamma}} \}^{-2}, \tag{52}$$

and that under the event $\mathcal{E}_B$, we have $T_B^Q = 0$. Then it holds that

$$\mathbb{P}\left\{ \left|v_t(B) - f(B)\right| > C_{\text{Lip}} r(B) + \mathbf{conf}_t(B) \right\}$$

$$= \mathbb{E}\left[ \mathbb{P}\left\{ \left|v_t(B) - f(B)\right| > C_{\text{Lip}} r(B) + \mathbf{conf}_t(B) \Big| \{(X_t^P, p_t^P)\}_{t=1}^{n_P} \right\} \right]$$

$$= \mathbb{E}\left[ \mathbb{P}\left\{ \left|v_t(B) - f(B)\right| > C_{\text{Lip}} r(B) + \mathbf{conf}_t(B) \Big| \{(X_t^P, p_t^P)\}_{t=1}^{n_P}, \mathcal{E}_B \right\} \mathbb{P}\left\{ \mathcal{E}_B \Big| \{(X_t^P, p_t^P)\}_{t=1}^{n_P} \right\} \right]$$

$$\quad + \mathbb{E}\left[ \mathbb{P}\left\{ \left|v_t(B) - f(B)\right| > C_{\text{Lip}} r(B) + \mathbf{conf}_t(B) \Big| \{(X_t^P, p_t^P)\}_{t=1}^{n_P}, \mathcal{E}_B^c \right\} \mathbb{P}\left\{ \mathcal{E}_B^c \Big| \{(X_t^P, p_t^P)\}_{t=1}^{n_P} \right\} \right]$$

$$\leq \mathbb{E}\left[ \mathbb{P}\left\{ \left|v_t(B) - f(B)\right| > C_{\text{Lip}} r(B) + \mathbf{conf}_t(B) \Big| \{(X_t^P, p_t^P)\}_{t=1}^{n_P}, \mathcal{E}_B \right\} \right] + \mathbb{P}\{\mathcal{E}_B^c\}$$

$$\leq 3\{ (\kappa n_P)^{\frac{d+3}{d+3+\gamma}} \}^{-2} \tag{53}$$

where the second inequality follows from (49) and (52).

**Case 3.** In this case, we consider $n_Q < (\kappa n_P)^{\frac{d+3}{d+3+\gamma}}$, and

$$\tilde{r} < \left\{ \frac{8 \log \left\{ (\kappa n_p)^{\frac{d+3}{d+3+\gamma}} \right\}}{C_{\mathcal{E}} \kappa n_P} \right\}^{\frac{1}{d+3+\gamma}}.$$

Then, we have that

$$8n_Q^{-2} T_B^Q + 2\{n_Q \vee (\kappa n_P)^{\frac{d+3}{d+3+\gamma}}\}^{-2}$$

$$\leq 8n_Q^{-2} \omega(B) + 2\{(\kappa n_P)^{\frac{d+3}{d+3+\gamma}}\}^{-2}$$

$$\leq 8n_Q^{-2} r(B)^{-2} \log \left\{ (\kappa n_p)^{\frac{d+3}{d+3+\gamma}} \right\} + 8n_Q^{-2} + 2\{(\kappa n_P)^{\frac{d+3}{d+3+\gamma}}\}^{-2}$$

$$\leq 16n_Q^{-2} \tilde{r}^{-2} \log \left\{ (\kappa n_p)^{\frac{d+3}{d+3+\gamma}} \right\} + 2\{(\kappa n_P)^{\frac{d+3}{d+3+\gamma}}\}^{-2}$$

$$\leq 18n_Q^{-2} \tilde{r}^{-2} \log \left\{ (\kappa n_p)^{\frac{d+3}{d+3+\gamma}} \right\}, \tag{54}$$

where the first inequality follows from (11), the second inequality follows from (12), the third inequality follows from $r(B) \geq \tilde{r}$ and the last inequality follows from $n_Q < (\kappa n_P)^{\frac{d+3}{d+3+\gamma}}$. Then combining (49) and (54), we have that

$$\mathbb{P}\left\{ |v_t(B) - f(B)| > C_{\text{Lip}} r(B) + \mathbf{conf}_t(B) \right\}$$

$$= \mathbb{E}\left[ \mathbb{P}\left\{ |v_t(B) - f(B)| > C_{\text{Lip}} r(B) + \mathbf{conf}_t(B) \Big| \{(X_t^P, p_t^P)\}_{t=1}^{n_P} \right\} \right]$$

$$\leq 18n_Q^{-2} \tilde{r}^{-2} \log \left\{ (\kappa n_p)^{\frac{d+3}{d+3+\gamma}} \right\}. \tag{55}$$

Finally, combining (51), (53) and (55), we complete the proof. $\qquad \square$

**Lemma B.2.** *Let $\{X_i\}_{i \geq 1}$ be bounded martingale difference sequence with $X_i \in [-1, 1]$. Then for any $\delta_1, \delta_2 > 0$, and integers $T \geq 0$ and $n \geq 1$, it holds that*

$$\mathbb{P}\left\{ \exists 0 \leq t \leq T : \left| \sum_{i=1}^{n} X_i + \sum_{i=n+1}^{n+t} X_i \right| > U(t, n, \delta_1, \delta_2) \right\} \leq 8T\delta_1 + 2\delta_2,$$

*where*

$$U(t, n, \delta_1, \delta_2) = \begin{cases} \sqrt{2(n+t) \log(1/\delta_1)} & \text{if } t > 0, \\ \sqrt{2n \log(1/\delta_2)}, & \text{if } t = 0. \end{cases}$$

*Proof.* The proof here is a minor modification of Lemma 12 in Cai et al. (2024a). For completeness, we provide the details.

We start with the case $T = 0$. Following the Azuma–Hoeffding inequality (e.g. Corollary 2.20 in Wainwright, 2019), it holds that

$$\mathbb{P}\left\{ \left| \sum_{i=1}^{n} X_i \right| > \sqrt{2n \log(1/\delta_2)} \right\} \leq 2\delta_2. \tag{56}$$

We consider the case $T > 0$. Note that

$$\mathbb{P}\left\{ \exists 0 \leq t \leq T : \left| \sum_{i=1}^{n} X_i + \sum_{i=n+1}^{n+t} X_i \right| > U(t, n, \delta_1, \delta_2) \right\}$$

$$\leq \mathbb{P}\left\{ \left| \sum_{i=1}^{n} X_i \right| > \sqrt{2n \log(1/\delta_2)} \right\} + \mathbb{P}\left\{ \exists 0 < t \leq T : \left| \sum_{i=1}^{n+t} X_i \right| > \sqrt{2(n+t) \log(1/\delta_1)} \right\}$$

$$\leq 2\delta_2 + \mathbb{P}\left\{ \exists t \in [T] : \left| \sum_{i=1}^{n+t} X_i \right| > \sqrt{2(n+t) \log(1/\delta_1)} \right\}, \tag{57}$$

where the second inequality follows from a union bound argument and the final inequality follows from (56).

It remains to control the second term in (57). Note that for any $\delta > 0$ and $T' > 0$, by Azuma–Hoeffding inequality (e.g. Corollary 2.20 in Wainwright, 2019), we have that

$$\mathbb{P}\left\{\exists t \in [T']: \left|\sum_{i=1}^{n+t} X_i\right| > \delta\right\} \leq \sum_{t=1}^{T'} 2\exp\left\{-\frac{\delta^2}{2(n+t)}\right\} \leq 2T'\exp\left\{-\frac{\delta^2}{2(n+T')}\right\}. \tag{58}$$

Note that

$$\mathbb{P}\left\{\exists t \in [T]: \left|\sum_{i=1}^{n+t} X_i\right| > \sqrt{2(n+t)\log(1/\delta_1)}\right\}$$

$$\leq \sum_{j=0}^{\lfloor \log_2(T)\rfloor} \mathbb{P}\left\{\exists 2^j \leq t \leq 2^{j+1}: \left|\sum_{i=1}^{n+t} X_i\right| > \sqrt{2(n+t)\log(1/\delta_1)}\right\}$$

$$\leq \sum_{j=1}^{\lfloor \log_2(T)\rfloor} 2^{j+2}\exp\left\{-2\left(\frac{n+2^j}{n+2^{j+1}}\right)\log(1/\delta_1)\right\}$$

$$\leq \sum_{j=1}^{\lfloor \log_2(T)\rfloor} 2^{j+2}\delta_1 \leq 2^{\log_2(T)+3}\delta_1 = 8T\delta_1, \tag{59}$$

where the second inequality follows from (58) and the third inequality follows from the fact that $(n+2^j)/(n+2^{j+1}) \geq 1/2$ for any integers $n, j \geq 0$.

Combining (57) and (59), we complete the proof.

$\square$

**Lemma B.3.** *For any $t \in [n_Q]$,*
$$\mathcal{Z} \subset \cup_{B \in \mathcal{A}_t} \boldsymbol{dom}(B, \mathcal{A}_t). \tag{60}$$

*Moreover, for any two different balls $B_1, B_2 \in \mathcal{A}_{n_Q}$ with the same radius $r$, their centres are at a distance of at least $r$.*

*Proof.* Note that for any $t \in [n_Q]$,
$$\cup_{B \in \mathcal{A}_t} \boldsymbol{dom}(B, \mathcal{A}_t) = \cup_{B \in \mathcal{A}_t} B,$$

By construction, $\mathcal{Z} \subset \cup_{B \in \mathcal{A}_t} B$. This completes the proof of (60).

For the second part, consider two distinct balls $B_1, B_2 \in \mathcal{A}_{n_Q}$ with radius $r$. By construction, they cannot be activated at the same time $t \in [n_Q]$. Without loss of generality, let $B_1$ be activated at time $t$, with parent $B^{\mathrm{par}}$, and $B_2$ be activated earlier. Denote the centre of $B_1$ as $(x^\top, p)^\top$. We have that $(x^\top, p)^\top \in \mathrm{dom}(B^{\mathrm{par}}, \mathcal{A}_t \backslash \{B_1\})$. Since $r(B^{\mathrm{par}}) > r(B_2)$ and $B_2$ is activated earlier, we have that $(x^\top, p)^\top \notin B_2$, which establishes the required separation between $B_1$ and $B_2$, and completes the proof. $\square$

**Lemma B.4.** *For any $t \in [n_Q]$ and $B \in \mathcal{A}_t$, denote*

$$\mathcal{E}_B = \left\{n_B^P(\mathcal{D}^P) \geq C_{\mathcal{E}}(\kappa n_P)r(B)^{d+\gamma+1}\right\},$$

*where $C_{\mathcal{E}} = c_\gamma c_Q$. Under Assumption 2.2, it holds that*

$$\mathbb{P}\{\mathcal{E}_B\} \geq 1 - \exp\left\{-4^{-1}C_{\mathcal{E}}(\kappa n_P)\tilde{r}^{d+\gamma+1}\right\}. \tag{61}$$

*Furthermore, if $n_Q < (\kappa n_P)^{\frac{d+3}{d+3+\gamma}}$ and*

$$\tilde{r} \geq \left\{\frac{8\log\left\{(\kappa n_p)^{\frac{d+3}{d+3+\gamma}}\right\}}{C_{\mathcal{E}}\kappa n_P}\right\}^{\frac{1}{d+3+\gamma}}, \tag{62}$$

*we have that*

$$\mathbb{P}\{\mathcal{E}_B\} \geq 1 - \{(\kappa n_p)^{\frac{d+3}{d+3+\gamma}}\}^{-2}, \tag{63}$$

*and under the event $\mathcal{E}_B$,*

$$T_B^Q = 0, \tag{64}$$

*where $T_B^Q$ is defined in* (11).

*Proof.* Fix time $t$ and a ball $B \in \mathcal{A}_t$, and recall its centre as $c(B) = (x_B^\top, p_B)^\top$. By Step 8 of Algorithm 1, we have that $x_B \in \mathrm{supp}(Q_X)$. Note that $n_B^P(\mathcal{D}^P)$ is a sum of i.i.d. zero mean Bernoulli random variables with

$$
\begin{aligned}
\mathbb{E}\{n_B^P(\mathcal{D}^P)\} =& n_P \mu(B) = n_P \mu\Big([p_B - r(B), p_B + r(B)] \times B_{\mathcal{X}}\big(x_B, r(B)\big)\Big) \\
\geq& 2n_P \kappa r(B) P_X\big(B_{\mathcal{X}}(x_B, r(B))\big) \geq 2c_\gamma n_P \kappa r(B)^{\gamma+1} Q_X\big(B_{\mathcal{X}}(x_B, r(B))\big) \\
\geq& 2c_\gamma c_Q n_P \kappa r(B)^{d+\gamma+1} = 2C_{\mathcal{E}} \kappa n_P r(B)^{d+\gamma+1},
\end{aligned}
$$

where the first inequality follows from Definition 2.4, the second inequality follows from Definition 2.3 and the third inequality follows from Assumption 2.2. As a consequence, by Chernoff's bound, we have that

$$
\begin{aligned}
\mathbb{P}\{\mathcal{E}_B^c\} =& \mathbb{P}\Big\{n_B^P(\mathcal{D}^P) < C_{\mathcal{E}} n_P \kappa r(B)^{d+\gamma+1}\Big\} \\
\leq& \mathbb{P}\Big\{\mathbb{E}\{n_B^P(\mathcal{D}^P)\} - n_B^P(\mathcal{D}^P) > \mathbb{E}\{n_B^P(\mathcal{D}^P)\}/2\Big\} \\
\leq& \exp\Big\{-4^{-1}C_{\mathcal{E}} \kappa n_P r(B)^{d+\gamma+1}\Big\}.
\end{aligned}
$$

Since $r(B) \geq \tilde{r}$, it holds that

$$\mathbb{P}\{\mathcal{E}_B^c\} \leq \exp\Big\{-4^{-1}C_{\mathcal{E}} \kappa n_P \tilde{r}^{d+\gamma+1}\Big\}, \tag{65}$$

which completes the proof of (61).

Now we consider the case where

$$n_Q < (\kappa n_P)^{\frac{d+3}{d+3+\gamma}} \quad \text{and} \quad \tilde{r} \geq \left\{\frac{8 \log\left\{(\kappa n_p)^{\frac{d+3}{d+3+\gamma}}\right\}}{C_{\mathcal{E}} \kappa n_P}\right\}^{\frac{1}{d+3+\gamma}}.$$

Note that

$$r(B) \geq \tilde{r} \geq \left\{\frac{8 \log\left\{(\kappa n_p)^{\frac{d+3}{d+3+\gamma}}\right\}}{C_{\mathcal{E}} \kappa n_P}\right\}^{\frac{1}{d+3+\gamma}} \geq \left\{\frac{8 \log\left\{(\kappa n_p)^{\frac{d+3}{d+3+\gamma}}\right\}}{C_{\mathcal{E}} \kappa n_P}\right\}^{\frac{1}{d+1+\gamma}}. \tag{66}$$

Combining (65) and (66), it holds that

$$\mathbb{P}\{\mathcal{E}_B^c\} \leq \{(\kappa n_p)^{\frac{d+3}{d+3+\gamma}}\}^{-2},$$

which proves (63). Under the event $\mathcal{E}_B$, we have that

$$
\begin{aligned}
&\omega(B) - n_B^P(\mathcal{D}^P) \\
\leq& \frac{\log\left\{(\kappa n_p)^{\frac{d+3}{d+3+\gamma}}\right\}}{r(B)^2} + 1 - C_{\mathcal{E}} \kappa n_P r(B)^{d+\gamma+1} \\
\leq& \frac{\log\left\{(\kappa n_p)^{\frac{d+3}{d+3+\gamma}}\right\}}{\tilde{r}^2} + 1 - C_{\mathcal{E}} \kappa n_P \tilde{r}^{d+\gamma+1} \\
\leq& \left(\frac{C_{\mathcal{E}}}{8}\right)^{\frac{2}{d+3+\gamma}} (\kappa n_P)^{\frac{2}{d+3+\gamma}} \log^{\frac{d+1+\gamma}{d+3+\gamma}}\left\{(\kappa n_p)^{\frac{d+3}{d+3+\gamma}}\right\} + 1 \\
&- 8^{\frac{d+1+\gamma}{d+3+\gamma}} C_{\mathcal{E}}^{\frac{-2}{d+3+\gamma}} (\kappa n_P)^{\frac{2}{d+3+\gamma}} \log^{\frac{d+1+\gamma}{d+3+\gamma}}\left\{(\kappa n_p)^{\frac{d+3}{d+3+\gamma}}\right\} \\
\leq& 1 - 2^{1/2} C_{\mathcal{E}}^{\frac{-2}{d+3+\gamma}} (\kappa n_P)^{\frac{2}{d+3+\gamma}} \log^{\frac{d+1+\gamma}{d+3+\gamma}}\left\{(\kappa n_p)^{\frac{d+3}{d+3+\gamma}}\right\} < 0,
\end{aligned}
$$

where the first inequality follows from (12) and the event $\mathcal{E}_B$, the second inequality follows from $r(B) \geq \tilde{r}$ and the third inequality follows from (62). Thus by (11), we have under the event $\mathcal{E}_B$, $T_B^Q = 0$, which proves (64).

$\square$

## C. Proof of Theorem 4.4

The proof of Theorem 4.4 is in Appendix C.1 with all necessary auxiliary results in Appendix C.2.

### C.1. Proof of Theorem 4.4

*Proof.* Let $\tilde{r} \in (0, 1/2)$ be specified later and define

$$S_{\mathcal{X},\tilde{r}} = \left\{ x = (x_1, x_2, \ldots, x_d) \in \mathcal{X} \mid x_i = (k_i - 1/2)2\tilde{r}, \ k_i \in [\lfloor 1/(2\tilde{r}) \rfloor], \ \forall i \in [d] \right\},$$

with cardinality $\lfloor 1/(2\tilde{r}) \rfloor^d$. We define a grid set of covariates as

$$S_{\mathcal{X},\tilde{r}} = \left\{ x_1^*, \ldots, x_{\lfloor 1/(2\tilde{r}) \rfloor^d}^* \right\}.$$

Next, define the integer $m = \lfloor c_m/\tilde{r} \rfloor^d$ for s constant $0 < c_m < 1/2$. Using Varshamov-Gilbert bound (e.g. Lemma 2.9 in Tsybakov, 2009), we can find a set of well-separated vectors $\Omega_m = \{\omega^{(i)}\}_{i=0}^M \subset \{\pm 1\}^m$ such that

$$\log_2(M) \geq \frac{m}{8} \quad \text{and} \quad \rho(\omega^{(i)}, \omega^{(j)}) \geq \frac{m}{8}, \quad \forall 0 \leq i < j \leq M, \tag{67}$$

where $\rho(\omega, \omega') = |\{k \in [m] : \omega_k \neq \omega_k'\}|$ is the Hamming distance between $\omega$ and $\omega'$.

In the same spirit of $S_{\mathcal{X},\tilde{r}}$, we define the grid set for price as

$$S_{\mathcal{P},\tilde{r}} = \left\{ (k - 1/2)\tilde{r}, k \in [\lfloor 1/\tilde{r} \rfloor] \right\} = \left\{ p_1^*, \ldots, p_{\lfloor 1/\tilde{r} \rfloor}^* \right\}.$$

In the following proof, we first construct a collection $\mathcal{H}_{\Omega_m} = \{(Q_X^\omega, \mu^\omega, f^\omega) \mid \omega \in \Omega_m\}$ of probability distributions, where $Q_X^\omega$ is a probability distribution over $\mathcal{X}$ representing the covariate distribution in target data, $\mu^\omega$ is a probability distribution over $\mathcal{Z}$ representing the joint distribution of covariate-price pairs in source data, and $f^\omega \colon \mathcal{Z} \to [0,1]$ is the reward function for both target and source data. We then verify the constructed $\mathcal{H}_{\Omega_m} \subset \mathcal{I}(\gamma, c_\gamma, \kappa, C_{\mathrm{Lip}}, c_Q)$, from which the lower bound follows by applying Proposition C.1, which is Proposition 1 in Kpotufe & Martinet (2020).

**Step 1. Constructing the distribution collection** $\mathcal{H}_{\Omega_m} = \{(Q_X^\omega, \mu^\omega, f^\omega) \colon \omega \in \Omega_m\}$**.**

**Constructing the target covariate distribution.** Let the target covariate distribution be independent of $\omega$, with the density denoted by $q_X$, defined for any $x \in \mathcal{X}$ as follows

$$q_X(x) = \begin{cases} q_1, & \text{if } x \in \cup_{i=1}^m B_{\mathcal{X}}(x_i^*, \tilde{r}/4), \\ q_0, & \text{if } x \in \mathcal{X} \backslash \cup_{i=1}^m B_{\mathcal{X}}(x_i^*, \tilde{r}), \\ 0, & \text{otherwise,} \end{cases} \tag{68}$$

where

$$q_1 = \frac{c_Q \tilde{r}^d}{\mathrm{Leb}(B_{\mathcal{X}}(x_1^*, \tilde{r}/4))} \quad \text{and} \quad q_0 = \frac{1 - mc_Q \tilde{r}^d}{\mathrm{Leb}(\mathcal{X} \backslash \cup_{i=1}^m B_{\mathcal{X}}(x_i^*, \tilde{r}))},$$

with $c_Q > 0$ defined in Assumption 2.2.

**Constructing the source covariate-price pair distribution.** Let the distribution of the source covariate-price pairs be independent of $\omega$. The density of the source covariate, denoted by $p_X$, is defined for any $x \in \mathcal{X}$ as follows

$$p_X(x) = \begin{cases} c_\gamma \tilde{r}^\gamma q_1, & \text{if } x \in \cup_{i=1}^m B_{\mathcal{X}}(x_i^*, \tilde{r}/4), \\ \delta, & \text{if } x \in \cup_{i=1}^m B_{\mathcal{X}}(x_i^*, \tilde{r}) \backslash B_{\mathcal{X}}(x_i^*, \tilde{r}/2), \\ q_0, & \text{if } x \in \mathcal{X} \backslash \cup_{i=1}^m B_{\mathcal{X}}(x_i^*, \tilde{r}), \\ 0, & \text{otherwise,} \end{cases} \tag{69}$$

where

$$\delta = \frac{1 - c_\gamma \tilde{r}^\gamma q_1 \text{Leb}\big( \cup_{i=1}^m B_{\mathcal{X}}(x_i^*, \tilde{r}/4)\big) - q_0 \text{Leb}\big( \mathcal{X} \backslash \cup_{i=1}^m B_{\mathcal{X}}(x_i^*, \tilde{r})\big)}{\text{Leb}\big( \cup_{i=1}^m B_{\mathcal{X}}(x_i^*, \tilde{r})\backslash B_{\mathcal{X}}(x_i^*, \tilde{r}/2)\big)}.$$

Let $\tilde{p} \in S_{\mathcal{P}, \tilde{r}}$ be defined later. The density of the conditional distribution of the source price $p \in [0, 1]$ given the source covariate $x \in \mathcal{X}$ is defined as

$$p_P(p|x) = \begin{cases} \kappa, & \text{if } p \in [\tilde{p} - \tilde{r}/2, \tilde{p} + \tilde{r}/2], \\ \frac{1 - \tilde{r}\kappa}{1 - \tilde{r}}, & \text{otherwise.} \end{cases} \tag{70}$$

**Constructing the reward distributions.** For both target and source data, let the random reward, conditional on the covariate $x$ and price $p$, be a Bernoulli random variable with parameter $f^\omega(x, p)$. The reward functions $\{f^\omega\}_{\omega \in \Omega_m}$ are constructed as follow. First, define the function $\phi \colon \mathbb{R}_+ \mapsto [0, 1]$ by

$$\phi(z) = \begin{cases} 1, & \text{if } 0 \le z < 1/4, \\ 2 - 4z, & \text{if } 1/4 \le z < 1/2, \\ 0, & \text{otherwise.} \end{cases}$$

Next, define the function $\varphi \colon \mathcal{Z} \mapsto [0, 1/4]$ via

$$\varphi(x, p) = C_\varphi \tilde{r} \phi\big( \|(x^\top, p)^\top\|_\infty / \tilde{r}\big),$$

where $C_\varphi = (C_{\text{Lip}} \wedge 1)/4$. For any $\omega \in \Omega_m$, the reward function $f^\omega \colon \mathcal{Z} \mapsto [0, 1]$ is defined as follows

$$f^\omega(x, p) = 1/2 + \sum_{i=1}^m \omega_i \varphi\big( x - x_i^*, p - \tilde{p}\big) \mathbb{1}\big\{ x \in B_{\mathcal{X}}(x_i^*, \tilde{r})\big\} \tag{71}$$

By construction, we note that if $x \in B_{\mathcal{X}}(x_i^*, \tilde{r}/4)$ for some $i \in [m]$, then $f^\omega(x, p)$ only depends on the value of $p$ and $\omega$; and if $x \in \mathcal{X} \setminus \bigcup_{i=1}^m B_{\mathcal{X}}(x_i^*, \tilde{r}/2)$, $f^\omega(x, p) = 1/2$ for any $p$ and $\omega$.

For any $\omega \in \Omega_m$ and $x \in \mathcal{X}$, let $p^{\omega,*}(x) = \min \big\{ p' \in \arg\max_{p \in [0,1]} f^\omega(x, p)\big\}$ be the optimal price. Furthermore, recall that $\tilde{p} \in S_{\mathcal{P}, \tilde{r}}$, we define the function $h \colon [0, 1] \to \{0, 1\}$ as

$$h(p) = \mathbb{1}\Big\{ p \in (\tilde{p} - \tilde{r}/2, \tilde{p} + \tilde{r}/2]\Big\}.$$

We now make the following remarks:

- For any $\omega \in \Omega_m$, $x \in \bigcup_{i=1}^m B_{\mathcal{X}}(x_i^*, \tilde{r}/4)$ and $p \in \bigcup_{i=1}^{\lfloor 1/\tilde{r} \rfloor} [p_i^* - \tilde{r}/4, p_i^* + \tilde{r}/4]$,

$$f^\omega\big(x, p^{\omega,*}(x)\big) - f^\omega(x, p) = C_\varphi \tilde{r} \mathbb{1}\big\{ h\big(p^{\omega,*}(x)\big) \neq h(p)\big\}. \tag{72}$$

  To see this, we first note that $x \in B_{\mathcal{X}}(x_i^*, \tilde{r}/4)$ for some $i \in [m]$. If $\omega_i = -1$, we have $p^{\omega,*}(x) = 0 \notin (\tilde{p} - \tilde{r}/2, \tilde{p} + \tilde{r}/2]$, and $h(p^{\omega,*}(x)) = 0$. Therefore, $f^\omega\big(x, p^{\omega,*}(x)\big) - f^\omega(x, p) = C_\varphi \tilde{r} \mathbb{1}\big\{ |p - \tilde{p}| \le \tilde{r}/2\big\} = C_\varphi \tilde{r} h(p) = C_\varphi \tilde{r} \mathbb{1}\big\{ h\big(p^{\omega,*}(x)\big) \neq h(p)\big\}$ where the first equality holds since $p \in \bigcup_{i=1}^{\lfloor 1/\tilde{r} \rfloor} [p_i^* - \tilde{r}/4, p_i^* + \tilde{r}/4]$. Next, if $\omega_i = 1$, we have $p^{\omega,*}(x) = \tilde{p} - \tilde{r}/4$, and $h(p^{\omega,*}(x)) = 1$. Hence $f^\omega\big(x, p^{\omega,*}(x)\big) - f^\omega(x, p) = C_\varphi \tilde{r}[1 - \mathbb{1}\big\{ |p - \tilde{p}| \le \tilde{r}/2\big\}] = C_\varphi \tilde{r}[1 - h(p)] = C_\varphi \tilde{r} \mathbb{1}\big\{ h\big(p^{\omega,*}(x)\big) \neq h(p)\big\}$.

- For any $\omega \in \Omega_m$, $p \in [0, 1]$, if $x \in \mathcal{X} \setminus \bigcup_{i=1}^m B_{\mathcal{X}}(x_i^*, \tilde{r}/2)$,

$$f^\omega(x, p) = 1/2. \tag{73}$$

- For any $\omega \in \Omega_m$ and $x \in \mathcal{X}$, if $p \in [0, 1] \backslash [\tilde{p} - \tilde{r}/2, \tilde{p} + \tilde{r}/2]$,

$$f^\omega(x, p) = 1/2. \tag{74}$$

- For any two different $\omega \neq \omega' \in \Omega_m$, if $x \in B_{\mathcal{X}}(x_i^*, \tilde{r}/4)$ for some $i \in [m]$ such that $\omega_i \neq \omega_i'$,

$$h\big(p^{\omega,*}(x)\big) \neq h\big(p^{\omega',*}(x)\big). \tag{75}$$

**Step 2. Verifying $\mathcal{H}_{\Omega_m} \subset \mathcal{I}(\gamma, c_\gamma, \kappa, C_{\text{Lip}}, c_Q)$.**

**Transfer exponent.** Recall that the support of $Q_X$ is the union of the set $\cup_{i=1}^m B_{\mathcal{X}}(x_i^*, \tilde{r}/4)$ and $\mathcal{X} \setminus \cup_{i=1}^m B_{\mathcal{X}}(x_i^*, \tilde{r})$. Therefore, we only need to check Definition 2.3 for $x$ in these sets.

First, if $x \in B_{\mathcal{X}}(x_i^*, \tilde{r}/4)$ for some $i \in [m]$, and $r \leq 3\tilde{r}/4$, by (68), it holds that

$$Q_X\big(B_{\mathcal{X}}(x, r)\big) = q_1 \text{Leb}\big(B_{\mathcal{X}}(x, r) \cap B_{\mathcal{X}}(x_i^*, \tilde{r}/4)\big).$$

By (69), it follows that

$$\begin{aligned} P_X\big(B_{\mathcal{X}}(x, r)\big) &\geq c_\gamma \tilde{r}^\gamma q_1 \text{Leb}\big(B_{\mathcal{X}}(x, r) \cap B_{\mathcal{X}}(x_i^*, \tilde{r}/4)\big) \\ &\geq c_\gamma r^\gamma Q_X\big(B_{\mathcal{X}}(x, r)\big). \end{aligned}$$

Second, for any $x \in \mathcal{X} \setminus \cup_{i=1}^m B_{\mathcal{X}}(x_i^*, \tilde{r})$ and $r \leq 3\tilde{r}/4$, by (68), it holds that

$$Q_X\big(B_{\mathcal{X}}(x, r)\big) = q_0 \text{Leb}\big\{B_{\mathcal{X}}(x, r) \cap \big(\mathcal{X} \setminus \cup_{i=1}^m B_{\mathcal{X}}(x_i^*, \tilde{r})\big)\big\}.$$

By (69), it holds that

$$P_X\big(B_{\mathcal{X}}(x, r)\big) \geq q_0 \text{Leb}\big\{B_{\mathcal{X}}(x, r) \cap \big(\mathcal{X} \setminus \cup_{i=1}^m B_{\mathcal{X}}(x_i^*, \tilde{r})\big)\big\} = Q_X\big(B_{\mathcal{X}}(x, r)\big).$$

Thus, for small $r \leq 3\tilde{r}/4$, the transfer exponent defined in Definition 2.3 equals $\gamma$ with respect to the constant $0 < c_\gamma \leq 1$.

Furthermore, since we set

$$P_X\big(B_{\mathcal{X}}(x_i^*, \tilde{r})\big) = Q_X\big(B_{\mathcal{X}}(x_i^*, \tilde{r})\big), \quad \forall i \in [m],$$

we can verify that the above inequalities hold for $3\tilde{r}/4 < r \leq 1$. Thus, the transfer exponent of the constructed source covariate distribution with respect to the target covariate distribution is $\gamma$, with the corresponding constant $c_\gamma$.

**Exploration coefficient.** For any $x \in \mathcal{X}$, $r \in (0, 1]$ and $p \in [r, 1 - r]$, we have that

$$\mu\big([p - r, p + r] \times B_{\mathcal{X}}(x, r)\big) \geq P_X\big(B_{\mathcal{X}}(x, r)\big) 2r \min\left\{\kappa, \frac{1 - \tilde{r}\kappa}{1 - \tilde{r}}\right\} = P_X\big(B_{\mathcal{X}}(x, r)\big) 2r\kappa,$$

where the first inequality follows from (70) and the final equality holds beacuse

$$\kappa - \frac{1 - \tilde{r}\kappa}{1 - \tilde{r}} = \frac{\kappa - 1}{1 - \tilde{r}} \leq 0.$$

As a result, the exploration coefficient defined in Definition 2.4 for the constructed source covariate-price pair distribution equals $\kappa$.

**Lipschitz condition.** By the construction in (71), it suffices to show that the function $\varphi \colon \mathcal{Z} \mapsto [0, 1/4]$ satisfies Assumption 2.1 with respect to the Lipschitz constant $C_{\text{Lip}} > 0$. For any $(x, p), (x', p') \in \mathcal{Z}$, we have that

$$|\varphi(x, p) - \varphi(x', p')| = C_\varphi \tilde{r}\big|\phi(\|(x^\top, p)^\top\|_\infty/\tilde{r}) - \phi(\|(x'^\top, p')^\top\|_\infty/\tilde{r})\big| \tag{76}$$

Since $\phi$ is a Lipschitz function with Lipschitz constant 4, it follows that

$$\begin{aligned} |\varphi(x, p) - \varphi(x', p')| &\leq 4C_\varphi \tilde{r}\big|\|(x^\top, p)^\top\|_\infty/\tilde{r} - \|(x'^\top, p')^\top\|_\infty/\tilde{r}\big| \\ &\leq 4C_\varphi\|(x^\top, p)^\top - (x'^\top, p')^\top\|_\infty \leq C_{\text{Lip}}\|(x^\top, p)^\top - (x'^\top, p')^\top\|_\infty, \end{aligned}$$

where the first inequality follows from the triangle inequality and the last inequality holds since $C_\varphi = (C_{\text{Lip}}/4) \wedge (1/4)$. Thus, for any $\omega \in \Omega_m$, the reward function $f^\omega$ satisfy Assumption 2.1 with respect to the Lipschitz constant $C_{\text{Lip}} > 0$.

**Lower-bounded density.** It is straightforward to see that

$$q_1 = \frac{c_Q \tilde{r}^d}{(\tilde{r}/2)^d} = 2^d c_Q,$$

and

$$q_0 = \frac{1 - m c_Q \tilde{r}^d}{1 - m(2\tilde{r})^d} = \frac{1 - \lfloor c_m/\tilde{r} \rfloor^d c_Q \tilde{r}^d}{1 - \lfloor c_m/\tilde{r} \rfloor^d (2\tilde{r})^d} = 1 + \frac{\lfloor c_m/\tilde{r} \rfloor^d \tilde{r}^d (2^d - c_Q)}{1 - \lfloor c_m/\tilde{r} \rfloor^d (2\tilde{r})^d} > 1.$$

Thus, we confirm that the constructed target covariate distribution satisfies Assumption 2.2.

**Step 3. Applying Proposition C.1**. We now verify the conditions required for applying Proposition C.1.

**Step 3.1. Verification of condition** $(i)$ **in Proposition C.1.** For any admissible price policy $\pi = \{p_1^\pi, \ldots, p_{n_Q}^\pi\}$ and for $\omega \in \Omega_m$, let

$$R_{\pi,\omega}(n_Q) = \sum_{t=1}^{n_Q} \mathbb{E}_\omega \big\{ f^\omega\big(X_t, p^{\omega,*}(X_t)\big) - f^\omega(X_t, p_t^\pi) \big\},$$

where $\mathbb{E}_\omega$ denotes the expectation under the distribution $(Q_X^\omega, \mu^\omega, f^\omega)$. Note that for any price policy $\pi$, if there exists a time point $t' \in [n_Q]$ such that

$$p_{t'}^\pi \in \bigcup_{i=1}^{\lfloor 1/\tilde{r} \rfloor} [p_i^* - \tilde{r}/2, p_i^* + \tilde{r}/2] \setminus [p_i^* - \tilde{r}/4, p_i^* + \tilde{r}/4],$$

then there exists a policy $\pi'$ with

$$p_{t'}^{\pi'} \in \bigcup_{i=1}^{\lfloor 1/\tilde{r} \rfloor} [p_i^* - \tilde{r}/4, p_i^* + \tilde{r}/4],$$

and $p_t^\pi = p_t^{\pi'}$ for all $t \neq t'$, which satisfies

$$R_{\pi',\omega}(n_Q) \leq R_{\pi,\omega}(n_Q), \quad \text{for any } \omega \in \Omega_m.$$

To see this, suppose $p_{t'}^\pi \in [p_i^* - \tilde{r}/2, p_i^* + \tilde{r}/2] \setminus [p_i^* - \tilde{r}/4, p_i^* + \tilde{r}/4]$, for some $i \in [\lfloor 1/\tilde{r} \rfloor]$. If $\omega_i = -1$, then one simply chooses $p_{t'}^{\pi'} \in [p_j^* - \tilde{r}/4, p_j^* + \tilde{r}/4]$ for some $j \neq i$; otherwise if $\omega_i = 1$, then one chooses the same index $j = i$. Furthermore, if $x \in \mathcal{X} \setminus \bigcup_{i=1}^m B_\mathcal{X}(x_i^*, \tilde{r})$, then $f^\omega(x, p) = 1/2$ is universal for all $p \in [0, 1]$ due to (73).

Therefore, it suffices to consider price policies where $p_t^\pi \in \bigcup_{i=1}^{\lfloor 1/\tilde{r} \rfloor} [p_i^* - \tilde{r}/4, p_i^* + \tilde{r}/4]$ for all $t \in [n_Q]$. By (72), we have that

$$R_{\pi,\omega}(n_Q) = C_\varphi \tilde{r} \sum_{t=1}^{n_Q} \mathbb{E}_\omega \Big[ \mathbb{1}\big\{ h\big(p^{\omega,*}(X_t)\big) \neq h(p_t^\pi) \big\} \mathbb{1}\big\{ X_t \in \bigcup_{i=1}^m B_\mathcal{X}(x_i^*, \tilde{r}/4) \big\} \Big]$$

$$= C_\varphi \tilde{r} \sum_{t=1}^{n_Q} \sum_{i=1}^m \mathbb{E}_\omega \Big[ \mathbb{1}\big\{ h\big(p^{\omega,*}(X_t)\big) \neq h(p_t^\pi) \big\} \mathbb{1}\big\{ X_t \in B_\mathcal{X}(x_i^*, \tilde{r}/4) \big\} \Big].$$

Let $\Pi$ denote the space of all price policies. For any $p^\pi, p^{\pi'} \in \Pi$, define the semi-metric

$$\bar{\rho}(p^\pi, p^{\pi'}) = C_\varphi \tilde{r} \sum_{t=1}^{n_Q} \sum_{i=1}^m \mathbb{E}_\omega \Big[ \mathbb{1}\big\{ h(p_t^{\pi'}) \neq h(p_t^\pi) \big\} \mathbb{1}\big\{ X_t \in B_\mathcal{X}(x_i^*, \tilde{r}/4) \big\} \Big].$$

For any $\omega \in \Omega_m$, denote $\tilde{p}^{\omega,*} = \{p^{\omega,*}(X_1), \ldots, p^{\omega,*}(X_{n_Q})\}$. For any $\omega, \omega' \in \Omega_m$ we have that

$$
\begin{aligned}
\bar{\rho}\big(\tilde{p}^{\omega,*}, \tilde{p}^{\omega',*}\big) =& C_\varphi \tilde{r} \sum_{t=1}^{n_Q} \sum_{i=1}^{m} \mathbb{E}_\omega \Big[ \mathbb{1}\big\{h\big(p^{\omega,*}(X_t)\big) \neq h\big(p^{\omega',*}(X_t)\big)\big\} \mathbb{1}\big\{X_t \in B_{\mathcal{X}}(x_i^*, \tilde{r}/4)\big\} \Big] \\
\overset{(a)}{\geq}& C_\varphi \tilde{r} \sum_{t=1}^{n_Q} \sum_{i=1}^{m} \mathbb{E}_\omega \Big[ \mathbb{1}\big\{X_t \in B_{\mathcal{X}}(x_i^*, \tilde{r}/4)\big\}\big) \mathbb{1}\big\{\omega_i \neq \omega_i'\big\} \Big] \\
\overset{(b)}{=}& C_\varphi c_Q n_Q \tilde{r}^{d+1} \rho(\omega, \omega') \overset{(c)}{\geq} C_\varphi c_Q n_Q \tilde{r}^{d+1} \frac{m}{8} = \frac{C_\varphi c_Q}{8} n_Q \tilde{r}^{d+1} \lfloor c_m/\tilde{r} \rfloor^d \geq C_\rho n_Q \tilde{r},
\end{aligned}
\tag{77}
$$

where (a) holds by (75), (b) follows from (68), (c) is derived from (67), and $C_\rho > 0$ is a constant only depending on constant $C_\varphi$, $c_Q$ and $c_m$.

**Step 3.2. Verification of condition $(ii)$ in Proposition C.1.**

Fix a policy $\pi$, for any $u \in [\lfloor 1/\tilde{r} \rfloor]$, let

$$
O_u = \sum_{t=1}^{n_Q} \mathbb{1}\{p_t^\pi \in [p_u^* - \tilde{r}/2, p_u^* + \tilde{r}/2]\}.
\tag{78}
$$

Since

$$
\sum_{i=1}^{M} \sum_{u=1}^{\lfloor 1/\tilde{r} \rfloor} \mathbb{E}_{\omega^{(i)}}(O_u) \leq M n_Q,
$$

it follows that for at least $\lfloor M/2 \rfloor$ elements in $\{\omega^{(i)}\}_{i=1}^{M}$, there must exist at least one index $u \in [\lfloor 1/\tilde{r} \rfloor]$ such that

$$
\mathbb{E}_{\omega^{(i)}}(O_u) \leq \frac{2 n_Q}{\lfloor 1/\tilde{r} \rfloor}.
\tag{79}
$$

Therefore, we can choose one such $u$ and without loss of generality, assume that $\{\omega^{(i)}\}_{i=1}^{\lfloor M/2 \rfloor}$ satisfy (79). Define $\tilde{p} = p_u^*$ to construct the density of the conditional distribution of the source price given the source covariate in (70) in **Step 1**.

Denote the target data and source data by $\mathcal{D}_{n_Q}^Q = \{X_t, p_t^\pi, Y_t\}_{t=1}^{n_Q}$, $\mathcal{D}^P = \{X_t^P, p_t^P, Y_t^P\}_{t=1}^{n_P}$, respectively. For each $\omega^{(i)} \in \Omega_m$, let $\Theta^i$ be the joint distribution of the random variables in $\mathcal{D}_{n_Q}^Q$ and $\mathcal{D}^P$ induced by $\big(Q_X^{\omega^{(i)}}, \mu^{\omega^{(i)}}, f^{\omega^{(i)}}\big)$, i.e. under $\Theta^i$:

- the target data $\mathcal{D}_{n_Q}^Q$ follow the target covariate distribution $Q_X$ with density $q_X$, the price policy $\pi$ is represented by its conditional density $\tilde{\pi}(\cdot|X_t, \mathcal{F}_{t-1})$ and rewards drawn from Bernoulli$\big(f^{\omega^{(i)}}(X_t, p_t^\pi)\big)$, whose likelihood is denoted by $\theta_{f^{\omega^{(i)}}}$;

- the source data $\mathcal{D}^P$ follow the source covariate distribution $P_X$ (the marginal from $\mu^{\omega^{(i)}}$) with density $p_X$, the source price distribution with conditional density denoted by $p_P^{\omega^{(i)}}(\cdot|X_t)$ and rewards drawn from Bernoulli$\big(f^{\omega^{(i)}}(X_t^P, p_t^P)\big)$ again with likelihood denoted by $\theta_{f^{\omega^{(i)}}}$.

We denote by $\theta^i$ the density corresponding to $\Theta^i$, then

$$
\begin{aligned}
\theta^i(D_{n_P}^P, D_{n_Q}^Q) =& \prod_{t=1}^{n_Q} q_X(X_t) \tilde{\pi}\big(p_t^\pi | X_t, \mathcal{F}_{t-1}\big) \theta_{f^{\omega^{(i)}}}\big(Y_t | X_t, p_t^\pi\big) \\
& \times \prod_{t=1}^{n_P} p_X(X_t^P) p_P\big(p_t^P | X_t^P\big) \theta_{f^{\omega^{(i)}}}\big(Y_t^P | X_t^P, p_t^P\big).
\end{aligned}
\tag{80}
$$

Note that the densities involving the policy $\tilde{\pi}\big(p_t^\pi\big|X_t, \mathcal{F}_{t-1}\big)$ are identical since $\pi$ is fixed. For any $i \in [\lfloor M/2 \rfloor]$, and $\omega^{(0)} \in \Omega_m$,

$$\log\left\{\frac{d\Theta^i}{d\Theta^0}(D_{n_P}^P, D_{n_Q}^Q)\right\} = \sum_{t=1}^{n_Q} \log\left\{\frac{\theta_{f^{\omega^{(i)}}}\big(Y_t\big|X_t, p_t^\pi\big)}{\theta_{f^{\omega^{(0)}}}\big(Y_t\big|X_t, p_t^\pi\big)}\right\} + \sum_{t=1}^{n_P} \log\left\{\frac{\theta_{f^{\omega^{(i)}}}\big(Y_t^P\big|X_t^P, p_t^P\big)}{\theta_{f^{\omega^{(0)}}}\big(Y_t^P\big|X_t^P, p_t^P\big)}\right\}.$$

Taking the expectation with respect to $\Theta^i$, we obtain

$$\begin{aligned}
\mathrm{KL}(\Theta^i, \Theta^0) =& \mathbb{E}_{\omega^{(i)}}\left[\sum_{t=1}^{n_Q} \log\left\{\frac{\theta_{f^{\omega^{(i)}}}\big(Y_t\big|X_t, p_t^\pi\big)}{\theta_{f^{\omega^{(0)}}}\big(Y_t\big|X_t, p_t^\pi\big)}\right\} + \sum_{t=1}^{n_P} \log\left\{\frac{\theta_{f^{\omega^{(i)}}}\big(Y_t^P\big|X_t^P, p_t^P\big)}{\theta_{f^{\omega^{(0)}}}\big(Y_t^P\big|X_t^P, p_t^P\big)}\right\}\right] \\
=& \mathbb{E}_{\omega^{(i)}}\left[\sum_{t=1}^{n_Q} \log\left\{\frac{\theta_{f^{\omega^{(i)}}}\big(Y_t\big|X_t, p_t^\pi\big)}{\theta_{f^{\omega^{(0)}}}\big(Y_t\big|X_t, p_t^\pi\big)}\right\}\right] + \mathbb{E}_{\omega^{(i)}}\left[\sum_{i=1}^{n_P} \log\left\{\frac{\theta_{f^{\omega^{(i)}}}\big(Y_t^P\big|X_t^P, p_t^P\big)}{\theta_{f^{\omega^{(0)}}}\big(Y_t^P\big|X_t^P, p_t^P\big)}\right\}\right] \\
=& \mathrm{KL}^Q + \mathrm{KL}^P.
\end{aligned} \tag{81}$$

Note that

$$\begin{aligned}
\mathrm{KL}^Q &= \int_{\mathcal{X}} \mathbb{E}_{\omega^{(i)}}\left[\sum_{t=1}^{n_Q} \log\left\{\frac{\theta_{f^{\omega^{(i)}}}\big(Y_t\big|x, p_t^\pi\big)}{\theta_{f^{\omega^{(0)}}}\big(Y_t\big|x, p_t^\pi\big)}\right\}\right] q_X(x)dx \\
&= \int_{\bigcup_{j=1}^m B_{\mathcal{X}}(x_j^*, \tilde{r}/4)} \mathbb{E}_{\omega^{(i)}}\left[\sum_{t=1}^{n_Q} \mathbb{1}\big\{p_t^\pi \in [p_u^* - \tilde{r}/2, p_u^* + \tilde{r}/2]\big\} \log\left\{\frac{\theta_{f^{\omega^{(i)}}}\big(Y_t\big|x, p_t^\pi\big)}{\theta_{f^{\omega^{(0)}}}\big(Y_t\big|x, p_t^\pi\big)}\right\}\right] q_X(x)dx \\
&\leq \sum_{j\in[m]:\,\omega_j^{(i)} \neq \omega_j^{(0)}} Q_X\big(B_{\mathcal{X}}(x_j^*, \tilde{r}/4)\big) \mathbb{E}_{\omega^{(i)}}(O_u) \mathrm{KL}\big(\mathrm{Bernoulli}(1/2 + C_\varphi\tilde{r}), \mathrm{Bernoulli}(1/2 - C_\varphi\tilde{r})\big) \\
&\leq 32C_\varphi^2\tilde{r}^2 \sum_{j\in[m]:\,\omega_j^{(i)} \neq \omega_j^{(0)}} Q_X\big(B_{\mathcal{X}}(x_j^*, \tilde{r}/4)\big) \mathbb{E}_{\omega^{(i)}}(O_u) \\
&\leq 64C_\varphi^2 c_Q \frac{n_Q}{\lfloor 1/\tilde{r} \rfloor} m\tilde{r}^{d+2} \leq C_Q n_Q \tilde{r}^{d+3} m,
\end{aligned} \tag{82}$$

where

- the second equality follows from (68), (73), (74) and $\tilde{p}(x) = p_u^*$,
- the first inequality follows from (71) and (78),
- the second inequality follows from Lemma 15 in Cai et al. (2024a),
- the third inequality follows from (68) and (79),
- and $C_Q > 0$ is a constant only depending on constants $C_\varphi$ and $c_Q$.

We also have that

$$\begin{aligned}
\mathrm{KL}^P &= \int_{\mathcal{X}} \mathbb{E}_{\omega^{(i)}}\left[\sum_{t=1}^{n_P} \log\left\{\frac{\theta_{f^{\omega^{(i)}}}\big(Y_t^P\big|x, p_t^P\big)}{\theta_{f^{\omega^{(0)}}}\big(Y_t^P\big|x, p_t^P\big)}\right\}\right] p_X(x)dx \\
&= \int_{\bigcup_{j=1}^m B_{\mathcal{X}}(x_j^*, \tilde{r}/4)} \mathbb{E}_{\omega^{(i)}}\left[\sum_{i=1}^{n_P} \mathbb{1}\big\{p_t^P \in [p_u^* - \tilde{r}/2, p_u^* + \tilde{r}/2]\big\} \log\left\{\frac{\theta_{f^{\omega^{(i)}}}\big(Y_t\big|x, p_t^P\big)}{\theta_{f^{\omega^{(0)}}}\big(Y_i\big|x, p_t^P\big)}\right\}\right] p_X(x)dx \\
&\leq \kappa n_P \tilde{r} \sum_{j\in[m]:\omega_j^{(i)} \neq \omega_j^{(0)}} P_X\big(B_{\mathcal{X}}(x_j^*, \tilde{r}/4)\big) \mathrm{KL}\big(\mathrm{Bernoulli}(1/2 + C_\varphi\tilde{r}), \mathrm{Bernoulli}(1/2 - C_\varphi\tilde{r})\big) \\
&\leq 32C_\varphi^2 n_P \kappa \tilde{r}^3 \sum_{j\in[m]:\omega_j^{(i)} \neq \omega_j^{(0)}} P_X\big(B_{\mathcal{X}}(x_i^*, \tilde{r}/4)\big) \\
&\leq 32C_\varphi^2 c_Q c_\gamma n_P \kappa m\tilde{r}^{d+\gamma+3} = C_P \kappa n_P \tilde{r}^{d+\gamma+3} m,
\end{aligned} \tag{83}$$

where

- the second equality follows from (69), (73) and (74),

- the first inequality follows from (70) and (71),

- the second inequality follows from Lemma 15 in Cai et al. (2024a),

- the third inequality follows from (69),

- and $C_P > 0$ is a constant only depending on constants $C_\varphi$, $c_Q$ and $c_\gamma$.

Combining (81), (82) and (83), it holds that

$$\lfloor M/2 \rfloor^{-1} \sum_{i=1}^{[\lfloor M/2 \rfloor]} \mathrm{KL}(\Theta^i, \Theta^0) \leq C_Q n_Q \tilde{r}^{d+3} m + C_P \kappa n_P \tilde{r}^{d+\gamma+3} m. \tag{84}$$

Now set

$$\tilde{r} = c_r \Big( n_Q + (\kappa n_P)^{\frac{d+3}{d+3+\gamma}} \Big)^{-\frac{1}{d+3}}, \tag{85}$$

where $c_r > 0$ is a sufficiently small constant. From (67), it holds that

$$\lfloor M/2 \rfloor^{-1} \sum_{i=1}^{[\lfloor M/2 \rfloor]} \mathrm{KL}(\Theta^i, \Theta^0) \leq \alpha \log_2(\lfloor M/2 \rfloor).$$

where $0 < \alpha < 1/8$.

**Step 3.3.** We have verified that all conditions of Proposition C.1 hold for the family $\{\Theta_i\}_{i=0}^{\lfloor M/2 \rfloor}$. By applying Proposition C.1, Markov's inequality, (77) and (85), we obtain that

$$\inf_{\pi} \sup_{I \in \mathcal{I}(\gamma, c_\gamma, \kappa, C_{\mathrm{Lip}}, c_Q)} R_{\pi, I}(n_Q) \geq c n_Q \big( n_Q + (\kappa n_P)^{\frac{d+3}{d+3+\gamma}} \big)^{-\frac{1}{d+3}},$$

where $c > 0$ is a constant only depending on constants $C_{\mathrm{Lip}}$, $c_\gamma$ and $c_Q$. This completes the proof.

$\square$

### C.2. Auxiliary results

For completeness, we include Proposition 1 from Kpotufe & Martinet (2020) below, as it is a key tool for establishing the minimax lower bound.

**Proposition C.1** (Proposition 1 in Kpotufe & Martinet (2020))**.** *Let* $\{\Theta_h\}_{h \in \mathcal{H}}$ *be a family of distributions indexed by a subset* $\mathcal{H}$ *of a semi-metric space* $(\mathcal{F}, \bar{\rho})$*. Assume that there exists* $h_0, \ldots, h_M \in \mathcal{H}$*, with* $M \geq 2$*, such that*

*(i)* $\bar{\rho}(h_i, h_j) \geq 2s > 0$*,* $\forall 0 \leq i < j \leq M$*,*

*(ii)* $\Theta_{h_i} \ll \Theta_{h_0}$*,* $\forall i \in [M]$ *and the average KL-divergence to* $\Theta_{h_0}$ *satisfies*

$$\frac{1}{M} \sum_{i=1}^{M} \mathrm{KL}(\Theta_{h_i}, \Theta_{h_0}) \leq \alpha \log M, \quad \text{where } 0 < \alpha < 1/8.$$

*Let* $Z \sim \Theta_h$ *and let* $\hat{h} \colon Z \to \mathcal{F}$ *denote any improper learner of* $h \in \mathcal{H}$*. It holds that*

$$\sup_{h \in \mathcal{H}} \Theta_h \left( \bar{\rho}\left( \hat{h}(Z), h \right) \geq s \right) \geq \frac{\sqrt{M}}{1 + \sqrt{M}} \left( 1 - 2\alpha - \sqrt{\frac{2\alpha}{\log(M)}} \right) \geq \frac{3 - 2\sqrt{2}}{8}.$$

# D. Additional details in Section 5

## D.1. Additional results in Section 5.1

**Scenario 1.** The simulation results for **Configuration 2** of **Scenario 1**, as described in Section 5.1, are presented in Figure 3.

**Scenario 2.** The simulation results for **Configurations 1** and **2** of **Scenario 2**, as described in Section 5.1, are presented in Figure 4 and Figure 5, respectively.

To assess robustness of TLDP, we conducted additional simulations under **Configuration 1** in **Scenario 1**, using a list of mis-specified values of tuning parameters $(\kappa, \gamma)$, while keeping the true values fixed at $\kappa = 0.6$ and $\gamma = 1$. The results, presented in Figure 6, indicate that our method remains robust under moderate misspecification of these parameters.

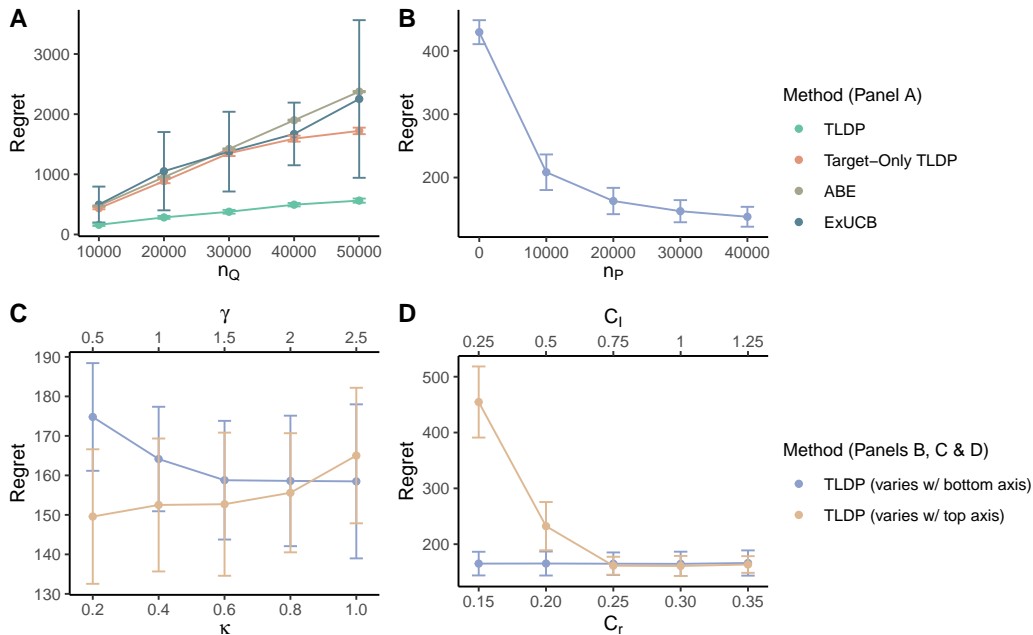

*Figure 3.* Results for Configuration 2 in Scenario 1. Panel (A) and (B): varying source data size $n_P$ and target data size $n_Q$, respectively. Panel (C) varying the transfer exponent $\gamma$ (top axis) and the exploration coefficient $\kappa$ (bottom axis). Panel (D): varying the index constant $C_I$ (top axis) and the exploration radius constant $C_r$ (bottom axis). For Panels (B), (C) and (D), we fix $n_Q = 10000$.

## D.2. Additional results in Section 5.2

In this subsection, we present samples of the processed auto loan dataset and the corresponding results in Tables 2 and 3, respectively.

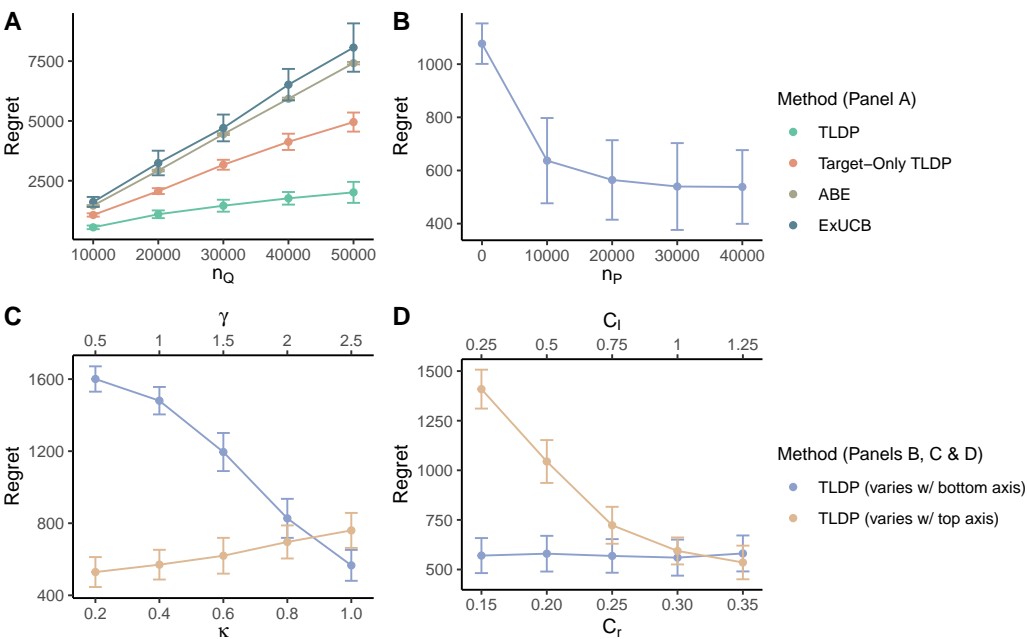

*Figure 4.* Results for Configuration 1 in Scenario 2. Panel (A) and (B): varying source data size $n_P$ and target data size $n_Q$, respectively. Panel (C) varying the transfer exponent $\gamma$ (top axis) and the exploration coefficient $\kappa$ (bottom axis). Panel (D): varying the index constant $C_I$ (top axis) and the exploration radius constant $C_r$ (bottom axis). For Panels (B), (C) and (D), we fix $n_Q = 10000$.

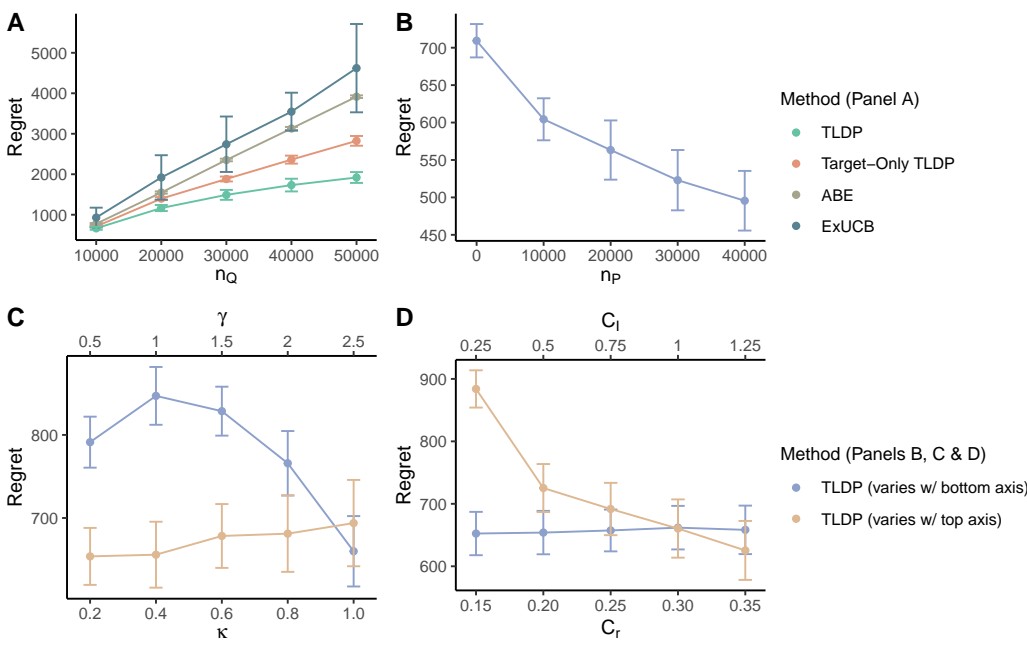

*Figure 5.* Results for Configuration 2 in Scenario 2. Panel (A) and (B): varying source data size $n_P$ and target data size $n_Q$, respectively. Panel (C) varying the transfer exponent $\gamma$ (top axis) and the exploration coefficient $\kappa$ (bottom axis). Panel (D): varying the index constant $C_I$ (top axis) and the exploration radius constant $C_r$ (bottom axis). For Panels (B), (C) and (D), we fix $n_Q = 10000$.

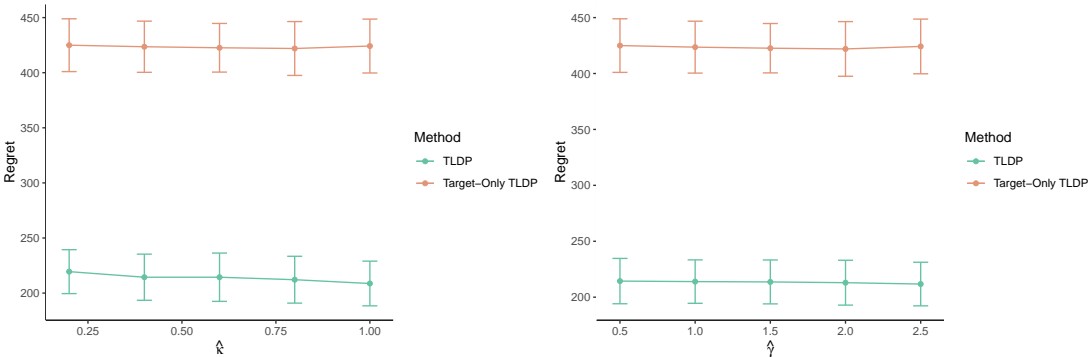

*Figure 6.* Results for Configuration 1 in Scenario 2. Left panel: Varying the estimated $\kappa$ with true fixed at $\kappa = 0.6$. Right panel: Varying the estimated $\gamma$ with true value fixed at $\gamma = 1$. In both panels, the target data size is fixed at $n_Q = 10000$ and the source data size at $n_P = 2n_Q$, and we set the index constant $C_I = 1$ and the exploration radius constant $C_r = 1/4$.

*Table 2.* Sample of the normalized auto loan dataset, including key covariates, prices and reward values.

| Division | Price | FICO | Competition rate | Amount approved | Prime rate | Term | Reward |
|---|---|---|---|---|---|---|---|
| West South Central | 0.26497654 | 0.4007937 | 0.9421965 | 0.31578947 | 0.7396567 | 1.0000000 | 0.00000000 |
| West South Central | 0.03455228 | 0.5912698 | 0.5664740 | 0.14842105 | 0.7396567 | 0.0000000 | 0.03455228 |
| East North Central | 0.08328724 | 0.7032520 | 0.8265896 | 0.2144407 | 0.7396567 | 0.3333333 | 0.00000000 |
| East North Central | 0.08203698 | 0.4227642 | 0.7687861 | 0.1620701 | 0.7396567 | 0.6666667 | 0.00000000 |
| East South Central | 0.09374806 | 0.2490119 | 0.9421965 | 0.03204972 | 0.7396567 | 1.0000000 | 0.00000000 |
| East South Central | 0.04983916 | 0.5731225 | 0.8265896 | 0.08330877 | 0.7396567 | 0.3333333 | 0.00000000 |

*Table 3.* Results for the auto loan dataset with East South Central data as the target division. Columns correspond to the source divisions utilized in TLDP. Here, $n_P$ represents the number of the source data utilized in TLDP and $n$ denotes the total number of source observations. Each cell reports the mean and standard deviation over 100 simulations.

| METHODS | MOUNTAIN | EAST NORTH CENTRAL | WEST SOUTH CENTRAL | PACIFIC |
|---|---|---|---|---|
| ABE | 70.81 (1.40) | 70.81 (1.40) | 70.81 (1.40) | 70.81 (1.40) |
| ExUCB | 64.91 (5.31) | 64.91 (5.31) | 64.91 (5.31) | 64.91 (5.31) |
| TLDP($n_P = 0$) | 71.95 (2.87) | 71.95 (2.87) | 71.95 (2.87) | 71.95 (2.87) |
| TLDP($n_P = 0.25n$) | 55.63 (8.04) | 56.80 (8.59) | 53.95 (8.26) | 51.61 (8.89) |
| TLDP($n_P = 0.5n$) | 54.29 (7.06) | 55.31 (8.60) | 54.07 (7.44) | 51.44 (7.12) |
| TLDP($n_P = 0.75n$) | 51.38 (6.92) | 51.52 (7.43) | 50.67 (7.19) | 49.40 (7.40) |
| TLDP($n_P = n$) | 50.23 (7.34) | 51.92 (7.09) | 49.85 (7.97) | 48.72 (8.27) |

