# OpenReview forum: "Transfer Learning for Nonparametric Contextual Dynamic Pricing"
_ICML.cc/2025/Conference — ICML 2025 poster_

### Official Review · Reviewer_uzaF · 2025-03-07

**Overall Recommendation:** 3

**Summary:**

The paper studies dynamic pricing problems using transfer learning techniques. The objective is to maximize expected total rewards and to minimize regret. The problem setup is stylized (monopoly scenario, time homogeneous demand), yet, reasonable. Numerical experiments are performed and results are compared to other baseline solutions. Overall, the paper is well-executed.

**Claims And Evidence:**

The claims of the paper appear valid and are supported by the numerical evaluation.

**Essential References Not Discussed:**

I do not miss essential references.

**Experimental Designs Or Analyses:**

The experimental design is reasonable. Small synthetic examples as well as real-world data based experiments are performed. Results are compared against other state-of-the-art baselines. Results are discussed and insights are inferred.

**Methods And Evaluation Criteria:**

The methods used are suitable and interesting. The regret criterion makes sense. The presentation of the concepts used is dense in some places but overall ok.

**Other Comments Or Suggestions:**

I suggest to better discuss limitations, scalability, and required parameter tuning of the approach.

**Other Strengths And Weaknesses:**

Strengths:

- The paper derives analytical results (theoretical bounds).

- The provided experiments are convincing.

Weaknesses:

- Limitations, scalability, and required parameter tuning of the approach could be better discussed.

**Questions For Authors:**

I have the following questions:

(1) Does the approach scale for larger d?

(2) What happens if the other product is not similar?

(3) Which hyperparameters have to be tuned?

(4) What are the limitations of the approach?

(5) Would it be possible to extend the setup to time-dependent demand and/or inventory considerations – as typical for e.g., airline ticket market, accommodation, etc.? Please discuss.


After reading the other reviews and the rebuttal, I decided to keep my score.

**Relation To Broader Scientific Literature:**

The literature using related methodologies is sufficiently discussed.

**Theoretical Claims:**

The theoretical claims and proofs appear solid. However, I did not check all proofs provided in the Appendix in full detail.

---

> ### Author Rebuttal · Authors · 2025-03-30
>
> We thank the reviewer for the detailed and helpful comments. The additional simulation results are given in this anonymized link. https://docs.google.com/document/d/e/2PACX-1vRBfGzJo3ETCltTWfOi_0p4RjLbsUJo6g9z0J-Ckm2m6fL0fahJWSrrptiFwOGCyxhtHNuyHsQP0tOh/pub
>
> **Q1** Our theoretical guarantees hold for any $d$.  Like most nonparametric works, however, our approach does encounter curse of dimensionality in high-dimensional settings, which can be seen from its regret upper bound. To mitigate this, one can impose additional structures on the reward function $f(x,p)$, where $x=(x^{(1)}, ..., x^{(d)})\in \mathbb R^d$ is the context. For example, one can adopt a sparse additive model $f(x,p)=\sum_{i=1}^d g_i(x^{(i)}, p)$, with sparsity $s \ll d$ (i.e. most $g_i(\cdot,\cdot)$ are zero functions). In this setting, the effective dimension is $s$ rather than $d$, albeit at the potential cost of model misspecification. Following your comment, we will provide a detailed discussion on this aspect.
>
> **Q2** First, we note that the context $X$ can encode useful information such as product specifications and consumer features, thus allowing the revenue function $f(x,p)$ to remain roughly stable across different products conditioned on a rich $X=x$. For example, in our real data experiments in Section 5.2, loans differ in their terms (e.g. duration). The source and target data differ in market conditions (i.e. different US regions). Despite these discrepancies, our method still improves performance, showing its robustness. Second, we further test robustness of our method under posterior drift (i.e. when revenue functions are different between source and target). Following the setup of Configuration 1 in Scenario 1, we modify the true source revenue function as $f(x, p) = \theta_0' + x^{\top} \theta' p + \tilde{\theta}' p^2,$ where $\theta_0' \sim \mathcal{N}(\theta_0, \sigma^2)$, $\theta' \sim \mathcal{N}(\theta, \sigma^2 I_d)$, and $\tilde{\theta}' \sim \mathcal{N}(\tilde{\theta}, \sigma^2)$, with $\theta_0, \theta, \tilde{\theta}$ being the target revenue function parameters. We set $\sigma = r \cdot \min(|\theta_0|, \|\theta\|_\infty, |\tilde{\theta}|)$ and control the drift severity by varying $r$. Results (Figure 1 in the link above) show our method performs well and is robust under moderate posterior drift.
>
> **Q3** One key hyperparameter is the smallest exploration radius $\tilde{r}$. While the regret bound remains valid for any choice of $\tilde{r}$ as discussed in Remark 4.2, the optimality of our TLDP algorithm relies on the specific $\tilde{r}$ given in equation (13). That choice depends on two parameters often unknown in practice: the transfer exponent $\gamma$ and the exploration coefficient $\kappa$.  In practice, $\kappa$ can be estimated from the source data, while $\gamma$ can be estimated by measuring the empirical overlap between the source and target covariate distributions using a small portion of the target data (e.g. $\sqrt{n_Q}$ samples). In our synthetic experiments, we used the true values of $\kappa$ and $\gamma$ to compute $\tilde{r}$. To assess robustness, we have conducted additional simulations under Configuration 1 in Scenario 1, using a list of mis-specified values of $(\kappa,\gamma)$, while keeping the true values fixed at $\kappa = 0.6$ and $\gamma = 1$. Results (Figure 2 in the link above) indicate that our method remains robust under moderate misspecification of these parameters.
>
> **Q4** As discussed in our response to your previous question, one limitation would be that to achieve optimal regret, our approach requires the knowledge of the unknown quantities $(\kappa, \gamma)$. It would be interesting to rigorously extend our method such that it is adaptive to the knowledge of $(\kappa, \gamma)$ and further establish theoretical guarantees for such an algorithm. In the revision, we will discuss more on this aspect of our limitation.
>
> **Q5** If the demand (thus revenue) function depends explicitly on time (e.g. seasonality, holiday, weekly effects), we can include time as an additional dimension in the context space. Our partition-based procedure then proceeds similarly and the theoretical guarantees still hold. However, if the revenue function $f(x,p)$ changes over time in a rather arbitrary fashion, we then need to rely on additional procedures such as change-point detection or moving-window estimation to counter the non-stationarity. We conjecture the regret upper bound will inevitably inflate in this case. As for inventory consideration, we conjecture one needs to reformulate the revenue maximization problem into a constrained optimization due to inventory limit. Based on deterministic approximation, Wang et al 2025 (on dynamic pricing with covariates) proposes a dual-based method for dynamic pricing under parametric models. We believe similar ideas can be used to help solve price $p_t$ in our Algorithm 1 under inventory constraint. We will comment on these aspects in the revision.

---

### Official Review · Reviewer_WLhP · 2025-03-11

**Overall Recommendation:** 3

**Summary:**

The authors study the problem of transfer learning in the context of dynamic pricing. Given a pre-collected source dataset, the authors propose an algorithm that exploits such a dataset to learn a partitioning of the joint context-price space in order to propose a price for each user with the goal of maximizing the cumulative revenue. The authors also study a lower bound of the problem and derive, as a consequence, both an upper and a lower bound in the case in which no source dataset is available.

---

Post rebuttal: The authors adequately answer my questions. Given that, I increased my score from 2 to 3.

**Claims And Evidence:**

The theorems proposed by the authors are supported by proofs. However, my concern is with respect to the assumptions required for the theorems to hold, as I will discuss later.

**Essential References Not Discussed:**

The references cited in this paper are sufficient to understand the motivation and the contributions of this work. To the best of my knowledge, no further references are necessary.

**Experimental Designs Or Analyses:**

I am not completely convinced by the validity of the experiments, but I am looking forward to the discussion with the authors to confirm or clear my concerns.

**Methods And Evaluation Criteria:**

The authors propose both a synthetic experiment and an experiment on real data. Regarding the synthetic experiment (i.e., Section 5.1), the simulation setting is crafted to match the assumptions required by the algorithm, which do not seem likely to hold in a real-world scenario, thus subtracting from the representativeness of the experiment. The experiment on real data (i.e., Section 5.2), on the other hand, seems to be much more promising, however some concerns remain, which will be discussed later.

**Other Comments Or Suggestions:**

Here I list some minor modifications that I believe could increase the readability of the work:

1. Line 57, right column, $\mathcal{Z} = \mathcal{X} \times \mathcal{P}$ is the more unambiguous definition of $\mathcal{Z}$;
2. The authors should formalize what is $Q_X (B)$ when $B$ is a ball (see Assumption 2.2);
3. Although understandable from the text, Algorithm 1 is demanding to read, and I would suggest the authors to write a higher-level algorithm in the main paper, reviewing and rewriting Algorithm 1 in the appendix;
4. In Theorem 4.1, it is not clear what the authors mean by writing "Assume that the target dataset, defined in (1)", as Equation (1) represents the expected reward.

**Other Strengths And Weaknesses:**

**Strengths**
This work in written in a clear and understandable way, guiding the reader through the authors' reasoning. Also, I believe the problem tackled by the authors to be of practical importance.

**Weaknesses**
My main concerns are on the problem formulation and the assumptions made in this work.
W1) The formulation of the random revenue seems to be incorrect. Although I understand the provenance of such a formulation, I believe it does not fit in the dynamic pricing setting, as the seller either observes a revenue of 0 (i.e., the customer does not buy the product) or of $p_t$ (i.e., the customer buys the product), and not a random revenue which can, in general, be in $[0,1]$. The simulation in Section 5.1 shows that a Gaussian is used to model the random revenue, which again does not seem correct.
W2) The assumption in Equation (3) (i.e., that the unknown reward functions are identical between the source problem and the target problem) seems to me as too strong, and unverifiable in the majority of applications. Such an assumption would require that every possible customer has the same propension to buy the two (in general, different) products for all the prices in the range. This is not in general true for different products and/or for different markets. The real-world experiment in Section 5.2 works well since the same product (i.e., loans) in the same market (i.e., the USA) is considered. However, I find this assumption very limiting.
W3) Considering weakness W2), it seems to me that the alternative setting (i.e., posterior drift) could have been a more applicable approach.

The weaknesses stated above have guided my judgement, however, in case the authors address them satisfactorily, I could reconsider my evaluation.

**Questions For Authors:**

I would like the authors to answer the following questions:
Q1) In Theorem 4.1, the authors write that the source dataset comprises "triplets independent across time". Does this mean that a dataset obtained from the observations collected in a different dynamic pricing application would not work, since each price would then be dependent on past observations? If yes, does removing such an assumption have an impact on the resulting regret bound?
Q2) In lines 264-266, the authors write "The choice of $\widetilde{r}$ in (13) ...". However, since $\widetilde{r}$ depends on $C_r$, and $C_r$ is, in turn, only defined in terms of two inequalities, is there a way to define a single optimal value of $C_r$?
Q3) From my understanding, $Q_X$ is a probability distribution, and as such $Q_X (\cdot) \in [0,1]$. Thus, following Assumption 2.2, I infer that $0 < c_Q < C_Q <= 1$. However, from the experiment in Section 5.1, at line 305, right column, a choice of $C_r = 1/4$ together with the conditions of line 247, left column, would result in $c_Q \ge 2048$. Can the authors clarify this point?
Q4) $P_X$ and $Q_X$ are not known in a real-world scenario, and $\gamma$ and $\kappa$ depend on these distributions. The value of $\widetilde{r}$, in turn, depends on $\gamma$ and $\kappa$. Given these observations and what the authors write in the conclusions, what could be the impact of misspecification, or of an interactive refinement, of these values on the regret? Moreover, what values have the authors considered for $\gamma$ and $\kappa$ in the experiment in Section 5.2?

Finally, I ask the authors to address my comments from the "Weaknesses" section of the review.

**Relation To Broader Scientific Literature:**

This work tackles an interesting problem, namely trying to exploit previous data to reduce the number of samples required to learn the pricing strategy of a new product, however, to the best of my understanding, some peculiarities of the specific setting have not been correctly addressed.

**Theoretical Claims:**

I have not thoroughly checked the proof in the appendix.

---

> ### Author Rebuttal · Authors · 2025-03-30
>
> We thank the reviewer for the detailed and helpful comments. The additional simulation results are given in this anonymized link. https://docs.google.com/document/d/e/2PACX-1vRBfGzJo3ETCltTWfOi_0p4RjLbsUJo6g9z0J-Ckm2m6fL0fahJWSrrptiFwOGCyxhtHNuyHsQP0tOh/pub
>
> **W1)** The classic $0$ or $p$ revenue model is a special case of our model. Formally, revenue $Y$ is $p\cdot D(x,p)$, where $D(x,p)$ is the random demand given context $x$ and price $p$ with $\mathbb E(D(x,p))=d(x,p)$. This leads to $\mathbb E(Y)=f(x, p) = p d(x, p)$. Thus we can directly model $f(x,p)$. When $D(x,p)$ is Bernoulli, it gives the 0 or $p$ revenue. However, demand $D(x,p)$ can be continuous (e.g. when sold by weight/volume like rice/gas). Also, we clarify in simulation scenario 2, we use [0,1]-truncated Gaussian centred at $f(x, p)$.
>
> **W2)&3)** Covariate shift is a widely used scheme in transfer learning (Suk&Kpotufe2021, Cai et al 2024).  The context $X$ can encode product specifications and consumer features, thus allowing revenue function to remain roughly stable across products or markets when conditioned on a rich $X$. In Section 5.2, loans differ in their terms (e.g. duration). The source and target data differ in market conditions (i.e. different US regions). Despite the discrepancies, our method still improves performance, showing its robustness.
>
> We further test robustness under posterior drift. Following the setup of Configuration 1 in Scenario 1, we modify the true source revenue function as $f(x, p) = \theta_0' + x^{\top} \theta' p + \tilde{\theta}' p^2,$ where $\theta_0' \sim \mathcal{N}(\theta_0, \sigma^2)$, $\theta' \sim \mathcal{N}(\theta, \sigma^2 I_d)$, and $\tilde{\theta}' \sim \mathcal{N}(\tilde{\theta}, \sigma^2)$, with $\theta_0, \theta, \tilde{\theta}$ being the target revenue parameters. We set $\sigma = r \cdot \min(|\theta_0|, \|\theta\|_\infty, |\tilde{\theta}|)$ and control the drift severity by varying $r$. Results (Figure 1 in the link above) show our method performs well and is robust under moderate posterior drift.
>
> **Q1)** The independence assumption (i.e. source data generated by a fixed behavior policy) is a standard assumption in bandit and RL literature, which simplifies theoretical analysis by enabling concentration inequalities for independent data. Our analysis can be extended to allow temporally dependent observations, such as those satisfying mixing conditions. Note that one must impose some structural conditions on the adaptivity of source data, as without them, data with arbitrary dependence can be uninformative. It would be interesting to allow source data to be collected via an adaptive policy. Doing so introduces additional complexities in terms of dependence structures and requires careful handling in the analysis to ensure valid results. We leave this for future research.
>
> **Q2)** The constant $C_r$ is used to define the smallest exploration radius $\tilde{r}$, which intuitively controls a bias-variance trade-off: $\tilde{r}$ small enough to yield fine local estimates v.s. large enough to avoid excessive partitioning and exploration. We do not have a single optimal value for $C_r$ in closed form as it depends on unknown quantities (e.g. $c_Q$ and $c_{\gamma}$). However, any constant $C_r$ in a range that satisfies the bounding inequalities leads to optimal regret (in order $O(\cdot)$). This is inline with bandit and RL literature, where one often settles for an order-wise optimal choice rather than a numerically unique constant.
>
> **Q3)** You are correct that $Q_X$ is a probability distribution, so Assumption 2.2 implies $c_Q\leq 1$. The constant 8 in $C_r^4c_{\gamma}c_Q\geq 8$ is merely sufficient for technical simplicity, and by no means necessary and sharp. While we conjecture that it may be technically possible to sharpen $8$ to some smaller constant to cover the choice of our experimental design, the magnitude of these constants does not affect the order of regret. In fact, our experiment shows our method is robust when the sufficient condition $C_r^4c_{\gamma}c_Q\geq 8$ is violated.
>
> **Q4)** Misspecification of $\gamma$ and $\kappa$ affects the optimal choice of $\tilde{r}$, and consequently, the regret bound, as discussed in Remark 4.2. In practice, $\kappa$ can be estimated from source data, while $\gamma$ can be estimated by measuring the empirical overlap between the source and target covariate distributions using a small portion of target data (e.g., $\sqrt{n_{Q}}$ samples). In our experiment, we use true $\kappa$ and $\gamma$ to compute $\tilde{r}$. To assess robustness, we conduct additional simulations under Configuration 1 in Scenario 1, using a list of mis-specified values of $(\kappa, \gamma)$, while fixing true values at $\kappa = 0.6$ and $\gamma = 1$. Results (Figure 2 in the link above) indicate our method remains robust under moderate misspecification of these parameters.
>
> We hope our responses help resolve your concerns and positively influence your evaluation.

---

> > ### Comment · Reviewer_WLhP · 2025-04-04
> >
> > I would like to thank the Authors for thoroughly addressing my questions and concerns.
> >
> > I appreciate the clarifications provided and the robustness of the experiments linked. As a final request, I would like the Authors to add such discussions (mainly those regarding the robustness) to the paper, as I have found them impactful in terms of practical applicability of this work.
> >
> > As a consequence, I am increasing my evaluation.

---

> > > ### Author Response · Authors · 2025-04-04
> > >
> > > Thank you very much for your acknowledgments, suggestions and appreciation.  We will endeavour to include the discussions, especially those on robustness, to the paper, within the length limit.

---

### Official Review · Reviewer_USgz · 2025-03-13

**Overall Recommendation:** 4

**Summary:**

The paper studies contextual dynamic pricing with nonparametric demands, a critical application in revenue management. The authors consider how transfer learning techniques can be applied for this problem, and achieve minimax optimal regret by devising a provably optimal online dynamic pricing algorithm while also providing matching information-theoretic lower bounds. They show the regret exhibits a phase transition phenomenon: when the source data size is small, the optimal regret rate is the same as the case where only target data is used, and the rate will benefit a lot from transfer learning when the source data size is larger. The authors also conduct numerical experiments to study the empirical behavior of their TLDP algorithm.

**Claims And Evidence:**

Yes. The paper's contributions are mainly on the theoretical side, and all assumptions are clearly stated and all claims are rigorously proved.

**Essential References Not Discussed:**

None.

**Experimental Designs Or Analyses:**

The experimental results look sound.

**Methods And Evaluation Criteria:**

The experiment section is easy to follow. The benchmark selection of the ABE and the ExUCB algorithms is reasonable.

**Other Comments Or Suggestions:**

None.

**Other Strengths And Weaknesses:**

The strength of the paper lies in the strong technical content involved in characterizing the minimax optimal regret for nonparametric contextual dynamic pricing with transfer learning. The analysis is quite skillful and the obtained results are strong.

In particular, the paper can be seen as an extension of [1], to a contextual bandit with continuous actions (specialized to pricing), since the major setup and assumptions are the same. A minor suggestion is that the authors need to comment on the additional technical challenges of analyzing the discretized version of the algorithm in [1] for the dynamic pricing setting (lack of such discussion is a current weakness of the paper).

[1] Cai, Changxiao, T. Tony Cai, and Hongzhe Li. "Transfer learning for contextual multi-armed bandits." The Annals of Statistics 52.1 (2024): 207-232.

**Questions For Authors:**

It is more common for researchers to study parametric demand models in dynamic pricing. Can you comment on why they solve the non-parametric problem first while not look into the parametric problem? In particular, how will your result look like or what are the challenges of characterizing the benefits of transfer learning for contextual dynamic pricing with (partially) linear demands (as in [1])?

[1] Bu, Jinzhi, David Simchi-Levi, and Chonghuan Wang. "Context-based dynamic pricing with partially linear demand model." Advances in Neural Information Processing Systems 35 (2022): 23780-23791.

**Relation To Broader Scientific Literature:**

The authors are the first to present a theoretical characterization of the benefits of applying transfer learning to the nonparametric contextual dynamic pricing setting.

**Theoretical Claims:**

I did not check the proofs carefully, but I did follow the main steps regarding the minimax optimality of regret. The proofs seem to be well-written.

---

> ### Author Rebuttal · Authors · 2025-03-31
>
> We thank the reviewer for the detailed and insightful comments.
>
> **Comparing our work with Cai et al (2024).** Cai et al (2024) study transfer learning for nonparametric contextual MAB under covariate shift. In their setting, actions (i.e. arms) are discrete and for simplicity are treated as a constant (i.e. the number of arms $K \asymp 1$).  The simplification $K \asymp 1$ results in their regret bound losing explicit control of $K$, making their approach inapplicable to infinite or uncountable action spaces. In contrast, dynamic pricing presents a unique challenge due to its continuous action space (i.e. prices), requiring more intricate methodologies than that in Cai et al (2024). Our approach effectively addresses this by adaptively partitioning the joint covariate-price space.  This adaptation ensures optimal regret scaling while accounting for both the covariates (of dimension $d$) and price (of dimension $1$). We will include the discussion in our revision.
>
>
> **Nonparametric assumption.** By not assuming a specific functional form for the demand curve, a nonparametric approach allows for more adaptability in capturing complex relationships between price and demand, especially when the true underlying demand structure is unknown or difficult to specify a priori. Therefore, compared with parametric model, the nonparametric model offers greater flexibility, which motivates our work.
>
> **Partially linear models in Bu et al (2022).** In terms of the partially linear model as you mentioned in Bu et al (2022), it covers two different demand models, $d(x,p)=bp+g(x)$ and $d(x,p)=f(p)+a^\top x$. For the first one, we believe that similar techniques developed in this paper, namely the adaptive exploration/partition of the covariate space, could be particularly useful for handling the nonparametric part $g(x)$ of the demand function and its transfer learning. On the other hand, due to the linearity in price, we conjecture a local linear demand model should be fitted within each bin to leverage the structure of the problem and achieve the optimal regret (instead of a local constant revenue function as in our paper). Since there is no interaction between $p$ and $x$ and there is linearity structure in $p$, we believe sharper rates than the ones in our paper can be achieved.
>
> For the second one, we conjecture a different upper bound strategy other than our adaptive exploration/partition should be used to leverage the linearity in $a^\top x$ and the smoothness in the one-dimensional function $f(p)$. In addition, for the linear part, under the covariate shift framework, we believe that techniques developed in existing literature, such as those in Lei et al (2021) and He et al (2024), can be applied for its transfer learning.
>
> A thorough investigation of transfer learning under the partially linear demand models is indeed an interesting avenue for future research. Following your suggestion, we will discuss these aspects in the revision.
>
> **References**
>
> [1] Cai, Changxiao, T. Tony Cai, and Hongzhe Li. "Transfer learning for contextual multi-armed bandits." The Annals of Statistics 52.1 (2024): 207-232.
>
> [2] Bu, Jinzhi, David Simchi-Levi, and Chonghuan Wang. "Context-based dynamic pricing with partially linear demand model." Advances in Neural Information Processing Systems 35 (2022): 23780-23791.
>
> [3] Lei, Qi, Wei Hu, and Jason Lee. Near-optimal linear regression under distribution shift. In International Conference on Machine Learning, pages 6164–6174. PMLR, 2021.
>
> [4] He, Zelin, Ying Sun, and Runze Li. Transfusion: Covariate-shift robust transfer learning for high-dimensional
> regression. In International Conference on Artificial Intelligence and Statistics, pages 703–711. PMLR, 2024

---

### Official Review · Reviewer_pvZ8 · 2025-03-16

**Overall Recommendation:** 3

**Summary:**

This paper introduces a novel Transfer Learning for Dynamic Pricing (TLDP) algorithm designed to effectively utilize pre-collected data from a source domain to improve pricing decisions in a target domain. The regret upper bound of TLDP is established under a straightforward Lipschitz condition on the reward function. To demonstrate the optimality of TLDP, we also derive a matching minimax lower bound, which encompasses the target-only scenario as a special case and is presented for the first time in the literature.

**Claims And Evidence:**

Yes

**Essential References Not Discussed:**

No

**Experimental Designs Or Analyses:**

Yes

**Methods And Evaluation Criteria:**

Yes

**Other Comments Or Suggestions:**

-

**Other Strengths And Weaknesses:**

1. The presentation is good. The authors clearly state their problem background as well as their contributions.
2. The paper proposes a novel algorithm for transfer learning in dynamic pricing , and provide regret upper/lower bounds of the problem, which is shown to be minimax optimal.

**Questions For Authors:**

-

**Relation To Broader Scientific Literature:**

This paper considers a transfer learning scenario in the dynamic pricing problem.

**Theoretical Claims:**

No

---

> ### Author Rebuttal · Authors · 2025-03-31
>
> Thank you for your appreciation, especially in acknowledging our presentation and our novelty.

---

### Decision · Program_Chairs · 2025-05-01

**Decision:**

Accept (poster)

**Comment:**

This work studies the contextual dynamic pricing problem with binary, unknown- and unparametric-distributional demands. Motivated by utilizing the sales history of other related products as a “warm start”, the authors consider a transfer learning framework for dynamic pricing on a covariate shift situation. Their TLDP algorithm achieves the minimax regret rate (as they also prove a matching lower bound), which also implies a two-stage phenomenon: as the source data size goes larger, the regret will switch from a cold-start rate ($n^{\frac{d+2}{d+3}}$) to a much smaller rate that benefits more from the source dataset. The authors conduct numerical experiments by comparing (and outperforming) TLDP with existing ABE and Explore-then-UCB algorithms, as well as demonstrating the influence of $n_p$, $\kappa$, and $C_r$ on the regret.

From the technical perspective, this work is an extension of Cai et al. (2024). Although the major setup and assumptions in this paper are the same as Cai et al. (2024), the extension to a contextual continuum-armed bandits is not that trivial. Given the overall quality and contributions, this paper is worth accepting.

However, there are still a few concerns that we kindly ask the reviewers to pay attention to:

(1) The assumption that the source and target demand functions are identical. This is not trivial nor granted even with covariates shifting being allowed. The authors should include a serious discussion with rigorous analysis to demonstrate the adaptivity (as you have conducted in your rebuttal but we expect a more comprehensive version).
(2) The $n^{\frac{d+2}{d+3}}$ regret bound, or a more general version like $n^{\frac{k+d+1}{2k+d+1}}$ for k-th smooth demands, is well-expected after Kleinberg (2004), Wang et al. (2021), and Cai and Pu (2022), as long as you are adapting the bandits setting to pricing. The authors are encouraged to consider an extension to a more general continuous family.

Reference:

Cai, Changxiao, T. Tony Cai, and Hongzhe Li. "Transfer learning for contextual multi-armed bandits." The Annals of Statistics 52.1 (2024): 207-232.
Kleinberg, Robert. "Nearly tight bounds for the continuum-armed bandit problem." Advances in Neural Information Processing Systems 17 (2004).
Wang, Yining, Boxiao Chen, and David Simchi-Levi. "Multimodal dynamic pricing." Management Science 67.10 (2021): 6136-6152.
Cai, T. Tony, and Hongming Pu. "Stochastic continuum-armed bandits with additive models: Minimax regrets and adaptive algorithm." The Annals of Statistics 50.4 (2022): 2179-2204.